



# Multi-year comparisons of ground-based and space-borne Fourier Transform Spectrometers in the high Arctic between 2006 and 2013

Debora Griffin[1], Kaley A. Walker[1,2], Stephanie Conway[1], Felicia Kolonjari[1], Kimberly Strong[1], Rebecca Batchelor[1,3], Chris D. Boone[2], Lin Dan[1], James R. Drummond[4], Pierre F. Fogal[1], Dejian Fu[2,5], Rodica Lindenmaier[1,6], Gloria L. Manney[7,8], and Dan Weaver[1]

[1]Department of Physics, University of Toronto, Toronto, Ontario, M5S 1A7, Canada
[2]Department of Chemistry, University of Waterloo, Waterloo, Ontario N2L 3G1, Canada
[3]Now at UCAR Center for Science Education, University Corporation for Atmospheric Research, Boulder, Colorado 80301, USA
[4]Department of Physics and Atmospheric Science, Dalhousie University, Halifax, Nova Scotia B3H 1Z9, Canada
[5]Now at Jet Propulsion Laboratory/California Institute of Technology, Pasadena, California 91109, USA
[6]Now at Pacific Northwest National Laboratory, Richland, Washington 99352, USA
[7]NorthWest Research Associates, Socorro, New Mexico, USA
[8]Department of Physics, New Mexico Institute of Mining and Technology, Socorro, New Mexico 87801, USA

*Correspondence to:* Kaley A. Walker
(kaley.walker@utoronto.ca)



**Abstract.** This paper presents eight years (2006-2013) of measurements obtained from Fourier Transform Spectrometers (FTSs) in the high Arctic at the Polar Environment Atmospheric Research Laboratory (PEARL, 80.05°N, 86.42°W). These measurements were taken as part of the Canadian Arctic ACE (Atmospheric Chemistry Experiment) Validation Campaigns that have been carried out since 2004 during the polar sunrise period (from mid-February to mid-April). Each spring, two ground-based FTSs were used to measure total and partial columns of HF, $O_3$, and trace gases that impact $O_3$ depletion, namely, HCl, and $HNO_3$. Additionally, some tropospheric greenhouse gases and pollutant species were measured, namely $CH_4$, $N_2O$, CO, and $C_2H_6$. During the same time period, the satellite-based ACE-FTS made measurements near Eureka and provided profiles of the same trace gases. Comparisons have been carried out between the measurements from PARIS-IR and the co-located high-resolution Bruker 125HR FTS, as well as with the latest version of the ACE-FTS retrievals (v3.5). The total column comparison between the two co-located ground-based FTSs, PARIS-IR and Bruker 125HR, found very good agreement for most of these species (except HF), with differences well below the estimated uncertainties ($\leq 6\%$) and with high correlations ($R \geq 0.8$). Partial columns have been used for the ground-based to space-borne comparison, with coincident measurements selected based on time, distance and scaled potential vorticity (sPV). The comparisons of the ground-based measurements with ACE-FTS show good agreement in the partial columns for most species within 6 % (except for $C_2H_6$ and PARIS-IR HF), which are consistent with the total retrieval uncertainty of the ground-based instruments. The correlation coefficients ($R$) of the partial column comparisons for all eight species range from approximately 0.75 to 0.95. The comparisons show no significant increase in the mean differences over these eight years, indicating the consistency of these datasets and suggesting that the space-borne ACE-FTS measurements have been stable over this period. In addition, changes in the amounts of these trace gases during springtime between 2006 and 2013 are presented and discussed. Increased $O_3$ ($0.9\,\%\mathrm{yr}^{-1}$), HCl ($1.7\,\%\mathrm{yr}^{-1}$), HF ($3.8\,\%\mathrm{yr}^{-1}$), $CH_4$ ($0.5\,\%\mathrm{yr}^{-1}$) and $C_2H_6$ ($2.3\,\%\mathrm{yr}^{-1}$, 2009-2013) have been found near PEARL from the Portable Atmospheric Research Interferometric Spectrometer for the InfraRed (PARIS-IR) dataset.

## 1 Introduction

Ground-based instruments provide valuable datasets for the validation of satellite-remote sensing instruments (e.g., Dils et al., 2014; Lacour et al., 2015). Regular validation of satellite instruments and their retrieval algorithms is necessary to assess the long-term stability of the measurements as well as the consistency of these datasets. As such, continuing validation of the space-borne Atmospheric Chemistry Experiment Fourier Transform Spectrometer (ACE-FTS) is essential to support its now over 10 year data record. ACE-FTS started routine measurements in February 2004, followed quickly by the first of the Canadian Arctic ACE Validation Campaigns which continue to this day. These campaigns (e.g., Kerzenmacher et al., 2005; Fraser et al., 2008; Batchelor et al., 2010; Fu et al., 2011; Adams et al., 2012) comprise ground-based measurements during the polar sunrise period (from the end of February to early April) at the Polar Environment Atmospheric Research Laboratory (PEARL) near Eureka, Nunavut (Fogal et al., 2013) at approximately 80° N 86° W. Two ground-based Fourier Transform Spectrometers (FTSs), the Portable Atmospheric Research Interferometric Spectrometer for the InfraRed (PARIS-IR), and the high-resolution Bruker 125HR FTS are part of these campaigns. Here, these datasets are used to compare multiple trace gas

species to the space-borne ACE-FTS v3.5 retrievals between 2006 and 2013 (Bernath et al., 2005) . These ground-based FTS datasets extend over a long time period and capture many species, contributing to the ongoing validation of the satellite-based instrument and help in assessing whether ACE-FTS measurements have remained consistent over the last decade.

These multi-year datasets can also help to quantify long-term changes in the Arctic tropospheric and stratospheric compo-
sition due to natural processes and anthropogenic emissions. Furthermore, the remote location of PEARL means there are few local pollutant sources, which helps in interpreting these changes in a global context without the influence of local contributions. The measurement period of these campaigns, i.e. the polar sunrise period, is of importance because it is a time during which chemical ozone depletion can occur. It is also a time that is dominated by highly variable dynamical conditions due to the polar vortex. The polar vortex is a large-scale cyclone (low pressure system) that extends from the upper troposphere to the
stratosphere. It forms in the winter and generally dissipates between late March and early April as the solar radiation increases (WMO, 2014). Trace gas amounts inside and outside the polar vortex are significantly different (e.g., Schoeberl et al., 1992). For a strong polar vortex, the vortex core is an isolated air mass and mixing with mid-latitude air only occurs around the outer edge. This leads to strong trace gas gradients across the edge of the polar vortex (e.g., Schoeberl et al., 1995; Manney et al., 1999). As such, it is important to consider the differences in the instrument viewing geometries with respect to the location
of the polar vortex when comparing the measurements. Employing a criterion ensuring that similar air masses are considered is crucial for instrument comparisons in the high Arctic, especially in the springtime when the polar vortex is at its strongest (Batchelor et al., 2010; Fu et al., 2011).

Herein, we focus on the retrieval of partial and total column values, derived from infrared FTS spectra, for $O_3$ together with several molecules important in catalytic $O_3$ destruction. These trace gases are $O_3$, HCl, and $HNO_3$ (e.g., Solomon, 1999). Also
retrieved is HF, which is a stratospheric tracer for dynamics (e.g., Toon et al., 1999). Additionally, total columns of primarily tropospheric $CH_4$, $N_2O$, CO, and $C_2H_6$ have been measured. Both $CH_4$, and $N_2O$ are useful dynamical tracers, as well as important greenhouse gases (e.g., IPCC, 2013). CO and $C_2H_6$ are both emitted from biomass burning, and are good tracers for long-range pollution transport (e.g., Yurganov et al., 2004; Griffin et al., 2013; Viatte et al., 2015, and references therein) due to their relatively long atmospheric lifetimes (on the order of months). All of these trace gases are routinely derived from
the Bruker 125HR spectra and have been used in numerous studies (e.g., Batchelor et al., 2009; Lindenmaier et al., 2010, 2011; Viatte et al., 2014; Holl et al., 2016). Retrievals of $O_3$, HCl, $HNO_3$, HF, $N_2O$, CO, and $C_2H_6$ from PARIS-IR's spectra have also been published and compared to other instruments in previous studies (e.g., Sung et al., 2007; Wunch et al., 2007; Batchelor et al., 2010; Fu et al., 2011; Griffin et al., 2013; Franco et al., 2016). The PARIS-IR $CH_4$ columns are presented for the first time in this study.

The earlier ACE-FTS retrieval version, ACE-FTS v2.2+updates, of these trace gases has previously been validated, and typical mean differences of partial columns between these and other ground-based FTSs in the Arctic are: $-9.1$ to $3.2\%$ for $O_3$ (Dupuy et al., 2009), 2.2 to $15.5\%$ for HCl (Mahieu et al., 2008), $-11.4$ to $2.4\%$ for $HNO_3$ (Wolff et al., 2008), 6.5 to $12.3\%$ for HF (Mahieu et al., 2008), 0.3 to $9.8\%$ for $CH_4$ (De Mazière et al., 2008), $-6.6$ to $3.8\%$ for $N_2O$ (Strong et al., 2008), and 15.6 to $28.9\%$ for CO (Clerbaux et al., 2008). In these studies, partial column comparisons between ground-
based FTSs and ACE-FTS in the Arctic typically show larger differences than comparisons at lower latitudes. The inclusion





of criteria ensuring that similar air masses are sampled with respect to the polar vortex reduces this difference significantly. Using these additional criteria, Batchelor et al. (2010) and Fu et al. (2011) found the following mean differences (using ACE-FTS v2.2+updates) for measurements with the Bruker 125HR and PARIS-IR between 2006 and 2008 near Eureka: $-5.2$ to $1.1$ % for $O_3$, $-4.6$ to $4.9\,\%$ for HCl, $-4.8$ to $5.7\,\%$ for $HNO_3$, and $-4.7$ to $5.2\,\%$ for HF. While these validation papers

have all used the previous ACE-FTS data version, this study focuses on the ACE-FTS v3.5 retrievals. The updates for v3.0/3.5 include new microwindows, updated spectroscopic parameters, and improved temperature and pressure retrievals (Boone et al., 2013). ACE-FTS v3.5 corrects for an error in a priori pressure and temperature profiles that impacted v3.0 in the period after September 2010 (Boone et al., 2013). Waymark et al. (2013) have shown general improvements between ACE-FTS v2.2 and ACE-FTS v3.0 across all baseline species ($O_3$, $H_2O$, $CH_4$, $N_2O$, $NO_2$, $NO$, $HNO_3$, $HCl$, $HF$, $CO$, $CCl_3F$, $CCl_2F_2$, $N_2O_5$,

and $ClONO_2$). Some studies have been published that compare ACE-FTS v3.0/3.5 and ground-based FTS retrievals including; $O_3$ and $NO_2$ by Adams et al. (2012), several carbon containing species (including $CO$ and $C_2H_6$) by Viatte et al. (2014), and $CH_4$ by Holl et al. (2016).

The aim of this study is to perform a detailed comparison between two ground-based FTSs and ACE-FTS over multiple years and also to assess changes in atmospheric composition above Eureka, utilizing eight years (2006-2013) of measurements during

the polar sunrise period. A comprehensive comparison of multiple trace gases is provided from ACE-FTS v3.5 and ground-based FTSs in Eureka including measurements that were taken inside and outside the polar vortex. The mean differences of the retrievals from the three FTSs throughout this time period as well as interannual changes in the total or partial column differences are determined. For these comparisons, we will use the same method and criteria for the viewing geometry as Batchelor et al. (2010) and Fu et al. (2011), which have been shown to reduce biases between ground- and satellite-based

instruments in the Arctic. Also, the interannual variability of the eight trace gases is discussed and changes in the trace gas columns near Eureka are investigated between 2006 and 2013.

This paper is organized as follows. Subsequent to this introduction, the measurement site, the instruments and the retrieval procedures used in this study are described. The third section discusses the comparison methodology and results of the ground-based intercomparisons between PARIS-IR and Bruker 125HR. The next section focuses on the methodology and results of

the ACE-FTS comparison results. The measurement series and trends from PARIS-IR measurements are presented in the fifth section. This is followed by the conclusions and highlights of our results.

## 2 Instrumentation and datasets

### 2.1 Measurement site

The ground-based measurements were taken at the Canadian Network for Detection of Atmospheric Change (CANDAC)

the PEARL Ridge Laboratory (80.05°N, 86.42°W, 610 m a.s.l.), in Eureka, Nunavut (Fogal et al., 2013). This laboratory is located 15 km away from the Eureka Weather Station (79.98° N, 85.93° W, 0 m a.s.l.) and over 400 km away from the closest settlement. This remote location minimizes the influence of locally polluted air on the atmospheric observations. It is also a



good location for measuring Arctic springtime ozone depletion, since the core of the polar vortex can be above Eureka, which is only $\sim 1100\,\mathrm{km}$ from the North Pole .

Ground-based measurements at PEARL have been carried out as part of the Canadian Arctic ACE Validation Campaigns during the polar sunrise period (typically from late-February to early-April) since 2004. As part of this campaign project,
two ground-based FTSs, PARIS-IR and the CANDAC Bruker 125HR, were operated simultaneously during the 2007-2013 campaigns to measure total as well as partial columns of the eight target species. These two instruments are located side-by-side in the PEARL Ridge Laboratory and share a solar beam from the same sun tracker installed on the roof above, where 1/3 of the beam is directed into PARIS-IR and 2/3 of the beam into the Bruker 125HR. During the campaigns, the satellite-based ACE-FTS took measurements near Eureka and provided profiles of over 30 trace gases. Details of these instruments and their
datasets are given below.

### 2.2 The Portable Atmospheric Research Interferometric Spectrometer for the InfraRed (PARIS-IR)

PARIS-IR is based on the design of the ACE-FTS (Fu et al., 2007). It was built by ABB Inc. in 2003 and has been part of the Canadian Arctic ACE Validation Campaigns since 2004. It records atmospheric solar absorption spectra between 750 and $4400\,\mathrm{cm}^{-1}$ at a maximum spectral resolution of $0.02\,\mathrm{cm}^{-1}$, equivalent to a maximum optical path difference (MOPD)
of $\pm 25\,\mathrm{cm}$. Since the 2006 campaign, the instrument has been operated in a consistent way and at its maximum spectral resolution. Interferograms are recorded using two liquid-nitrogen-cooled detectors, mercury cadmium telluride (HgCdTe) and indium antimonide (InSb) detectors, which are configured in a sandwich arrangement, and a zinc selenide (ZnSe) beam splitter. The entire spectral range ($750$–$4400\,\mathrm{cm}^{-1}$) is measured simultaneously for each observation, since no narrow-band filters are used. No apodization is applied to the spectra. Each recorded spectrum consists of 20 co-added spectra, taken approximately
every $7\,\mathrm{min}$ (Sung et al., 2007). All eight species of interest are measured every $7\,\mathrm{min}$ throughout the campaign period, whenever there are favourable weather conditions.

### 2.3 Bruker 125HR

The CANDAC Bruker 125HR is a high-resolution ground-based FTS, operated to produce atmospheric solar absorption spectra. During the sunlit period, it measures mid-infrared atmospheric solar absorption between 600 and $4300\,\mathrm{cm}^{-1}$ at a maximum
resolution of $0.0035\,\mathrm{cm}^{-1}$ (equivalent to a MOPD of $257\,\mathrm{cm}$) (Batchelor et al., 2009). It was installed at PEARL in July 2006 and is part of the Network for the Detection of Atmospheric Composition Change (NDACC, http://www.ndsc.ncep.noaa.gov/). These spectra are recorded with either a HgCdTe or InSb detector using a potassium bromide (KBr) beam splitter. Seven narrow-band filters are used and no apodization is applied to the spectra. During each campaign, spectra were recorded approximately every 4-8 min, and are comprised of either two or four co-added spectra. Therefore, subject to favourable weather
conditions and depending on the filter range, each species is measured approximately every $30\,\mathrm{min}$. All eight species of interest are retrieved from the Bruker 125HR spectra.





## 2.4 Ground-based retrieval algorithm

The same retrieval algorithm has been utilized to estimate the total column amounts of trace gases from the solar absorption spectra recorded by both ground-based FTSs. The retrieval technique applied is based on an optimal estimation method (OEM) (Rodgers, 1976, 2000). This is an iterative process, wherein a calculated spectrum is fitted to the observed one by adjusting the

target trace gas profile. Single or multiple microwindows, typically each with a width between 0.3 and $1.0 \, \mathrm{cm}^{-1}$, are employed in the retrieval process. Table 1 lists the microwindows used and interfering trace gases taken into account for the retrieval of each gas. These are consistent for both instruments and follow the recommendations from the InfraRed Working Group of NDACC (IRWG, http://www.acd.ucar.edu/irwg/).

Atmospheric profiles have been retrieved from the spectra with the SFIT4 version 0.9.4.4 retrieval package (https://wiki.ucar.

edu/display/sfit4/Infrared+Working+Group+Retrieval+Code,+SFIT) and the HIgh-resolution TRANsmission database (HITRAN) 2008 spectroscopic database (Rothman et al., 2009). Total column amounts were calculated within SFIT4 from the retrieved Volume Mixing Ratio (VMR) profiles. These columns are estimated by integrating the retrieved VMR profiles and the atmospheric density between the ground and the top of the atmosphere for the total columns, or over given altitude ranges specified for the partial columns. Due to the lower spectral resolution of PARIS-IR compared to the Bruker 125HR, two dif-

ferent altitude grids have been used for the retrieval. The retrievals from PARIS-IR spectra have been performed on a 29-layer grid (from the ground ($0.61 \, \mathrm{km}$) to $100 \, \mathrm{km}$) and those for the Bruker 125HR on a 47-layer grid (from the ground ($0.61 \, \mathrm{km}$) to $120 \, \mathrm{km}$). It has been shown using SFIT2 that this prevents non-physical oscillations in the retrieved profile for the lower resolution FTS, but only results in a very small difference in the total columns (between 0.1–0.6 % depending on the retrieved gas) (Wunch et al., 2007). We have confirmed that this is still valid for SFIT4 by testing the retrieval on both 29-layer and 47-layer

grids. For the retrieval of $\mathrm{CH_4}$ from the Bruker 125HR spectra, the retrieval strategy that incorporates a first-order Tikhonov constraint to the state vector as recommended by Sussmann et al. (2011) has been used to prevent non-physical oscillations of the retrieved profiles. This was not necessary for the PARIS-IR $\mathrm{CH_4}$ retrieval since no oscillations occurred in the retrieved profiles with the standard retrieval technique.

The retrieval algorithm requires input meteorological parameters, which are used in the radiative transfer calculation. Daily

pressure and temperature profiles (versus altitude) are calculated from National Centers for Environmental Prediction (NCEP, ftp://ftp.cpc.ncep.noaa.gov/ndacc/ncep/) profiles interpolated to PEARL and are used to approximately $1.0 \, \mathrm{mbar}$ ($\sim 45 \,$ km). Above this altitude, the monthly mean pressure and temperature profiles from the Whole Atmosphere Chemistry Community Model (WACCM, https://www2.acd.ucar.edu/gcm/waccm, Eyring et al., 2007) v6 for Eureka are used. The inversion procedure of the OEM requires a priori information for the gases involved in the retrieval (target and interfering species) to stabilize the

solution. This a priori knowledge refers to the VMR profile and its variability. The a priori profiles used for the retrieval of trace gases are the mean of a 40 yr run (1980–2020) of WACCM v6 for Eureka, as recommended by NDACC/IRWG. Only one single a priori profile for each species is used for the entire retrieval of the dataset for both FTSs. This provides consistency within the retrievals and ensures that variability in the dataset results from the measurements. A forward model within SFIT is





used to generate a model atmosphere from this a priori information based on the daily pressure and temperature information, as well as the location of the measurement site.

The characterization of the information content of the retrieval is the primary benefit of the OEM approach. The averaging kernel **A** is a matrix that characterizes this information (Rodgers, 2000). The total column averaging kernel (solid lines) and the

sensitivity (dashed-dotted lines) of the retrieval for each of the eight species for PARIS-IR (red) and the Bruker 125HR (blue) are shown in Fig. 1. The total column averaging kernel is estimated by the sum of the columns of **A** at each altitude, and the sensitivity equals the sum of the rows of **A** at each altitude. The sensitivity presents the fraction of the retrieved value that is derived from the measurement rather than the a priori for a given altitude (Vigouroux et al., 2007). A sensitivity of 1 indicates that 100 % of the information at this altitude results from the measurement. Note that in some cases no dashed-dotted line is

visible in the figure, since the sensitivity and total column averaging kernel are very similar and the difference cannot be seen. It follows from Fig. 1 that the $O_3$ retrieval is sensitive from the surface to approximately 40 km and 50 km for PARIS-IR and the Bruker 125HR, respectively. The retrieval of $O_3$ and the following species are primarily sensitive (with a sensitivity that is at least 0.1) in the stratosphere in the range given (and are, therefore, referred to in the following sections as "stratospheric species"): HCl from 10 km to 40 km (60 km for the Bruker 125HR); $HNO_3$ from 10 km to 40 km; and HF from 10 km to

40 km (50 km for the Bruker 125HR). Retrievals for the other species ($N_2O$, $CH_4$, CO, and $C_2H_6$) are mainly sensitive in the troposphere and lower stratosphere and are referred to in this study as "tropospheric species". The retrieved columns of $N_2O$ and $C_2H_6$ for both instruments are mainly sensitive from the surface to almost 30 km and almost 20 km, respectively. The CO and $CH_4$ retrievals from PARIS-IR are primarily sensitive between the surface and approximately 20-30 km, with significantly smaller sensitivity in the stratosphere than in the troposphere. The sensitivity for these species is different for the

retrievals from the Bruker 125HR, which are sensitive in the troposphere as well as in the stratosphere up to approximately 40 km and 80 km for $CH_4$ and CO, respectively. The altitudes for which the retrievals are most sensitive are later used (in Sect. 4) to determine the range of the partial columns for the comparison between the ground-based and satellite-borne instruments. Because a different retrieval technique has been used to determine the Bruker 125HR $CH_4$ (Sussmann et al., 2011), the total column averaging kernel is forced to 1 at all altitudes. The Degrees of Freedom for Signal (DOFS) are a measure of the vertical

information of the retrieved profile. It is defined as the trace of the averaging kernel matrix **A** (Rodgers, 2000). The DOFS of each retrieved species used in this study can be found in Table 1. For quality assurance, a RMS/DOFS filter (see Table 1) has been applied to the retrieved datasets, as presented in Sussmann et al. (2011).

The retrieval uncertainties are derived with SFIT4 by employing the method described by Rodgers (2000) and are listed in Table 1. The total uncertainties (given in Table 1) consist of the measurement error, the uncertainties of the line width and

line intensity parameters of the retrieved trace gas from HITRAN 2008 (Rothman et al., 2009) (where values are unavailable from HITRAN, 20 % has been used, see Table 2), and uncertainties caused by the temperature and Solar Zenith Angle (SZA) uncertainty (see Table 2). The measurement error is based on the signal-to-noise ratio of the observed spectra and determined by SFIT4, based on the algorithm described in Rodgers (1976) and Rodgers et al. (1990). The SZA error is based on the average change in the SZA during the time it takes to perform a measurement. The random and systematic temperature errors

(see Table 2) are based on comparison between averaged radiosondes and NCEP profiles, and the NCEP temperature error





profile, respectively. The total uncertainty has then been determined by adding all errors in quadrature. The smoothing error is not included in the total error (von Clarmann et al., 2014). The average total uncertainty of each species is listed in Table 1. The estimated uncertainty for $HNO_3$ is significantly larger than that of other species. This is due to the line intensity error and temperature broadening being unavailable in HITRAN 2008. Therefore, we used empirically estimated 20 % uncertainty for

$HNO_3$ line parameters in the measurement uncertainty analysis. This creates a large systematic error for $HNO_3$ and results in a total error of 19 %.

### 2.5 Atmospheric Chemistry Experiment Fourier Transform Spectrometer (ACE-FTS)

ACE-FTS was launched on board the Canadian satellite SCISAT on 12 August 2003. The satellite has a circular low-Earth orbit (650 km) with an inclination of $74°$ and therefore measurements cover tropical, midlatitude and polar regions over the

course of one year (Bernath et al., 2005). ACE is equipped with two instruments, ACE-FTS and the Measurement of Aerosol Extinction in the Stratosphere and Troposphere Retrieved by Occultation (MAESTRO) (McElroy et al., 2007). Its mission goals include improving our understanding of polar ozone chemistry and, thus every year during the Arctic sunrise period, ACE takes measurements over the high Arctic. The observation technique is solar occultation at sunrise and sunset. This work will focus on the FTS, which covers the spectral region between 750 and $4400\,cm^{-1}$ with a spectral resolution of $0.02\,cm^{-1}$

(which is identical to PARIS-IR). ACE-FTS has a vertical sampling of 1.5-6 km varying with the orbit and, based on its field-of-view, a vertical resolution of about 3–4 km (Boone et al., 2005). This is much higher than the vertical resolution of both ground-based FTSs, which typically retrieve partial or total columns with DOFS varying between 1 and 4.5 (see Table 1). The retrievals from ACE-FTS infrared spectra provide profiles for over 30 atmospheric trace gases as well as the meteorological variables of temperature and pressure (Boone et al., 2005).

In this study, the VMR as well as the temperature and pressure profiles are taken from the latest version, ACE-FTS v3.5 (Boone et al., 2013). The retrievals are based on the same global-fit method with a Levenberg-Marquardt nonlinear least squares fitting algorithm as used in the version 2.2+updates processing described by Boone et al. (2005). The range of the measurements is from the top of the clouds to the top of the atmosphere (at approximately 150 km). For clear-sky conditions the lower altitude range is approximately 5 km, depending on the season and location of the measurements.

### 2.6 Derived Meteorological Parameters

In the high Arctic, particularly during the highly-variable Arctic springtime, it is important to ensure that the air masses observed by the two instruments being compared are similar. Therefore, characterizing the viewing geometry with respect to polar vortex dynamics is essential to perform a robust instrument comparison in the high Arctic. We use the scaled Potential Vorticity (sPV; as calculated by Manney et al., 1994), as well as the temperatures along the line-of-sight for each instrument to

compare the similarity of the air masses sampled by the instrument with respect to the polar vortex. The ground-based line-of-sight calculations are described in Fu et al. (2011) and the sampling of the meteorological fields in Manney et al. (2007). The sPV profiles along the line-of-sight have been derived from GEOS version 5.2.0 analyses (GEOS-5) (Rienecker et al., 2008). This provides information on whether measurements were taken outside or inside the polar vortex. Within this study, the edge



of the polar vortex is defined to be between $1.2 \times 10^{-4}\,\mathrm{s}^{-1}$ and $1.6 \times 10^{-4}\,\mathrm{s}^{-1}$, which is consistent with the discussion in Manney et al. (2007) and with that used by Manney et al. (2008) and Batchelor et al. (2010).

## 3 Comparison between the two ground-based FTSs

### 3.1 Methodology

Throughout the multi-year Canadian Arctic ACE Validation Campaigns, efforts were made to provide the best possible conditions for instrument comparisons. Thus, as previously mentioned, the two ground-based FTSs made coincident measurements, sharing the same solar beam. During the Arctic sunrise period, the SZA is quite large (typically ranging between 75° and 90°) and consequently the locations of the air masses sampled by the FTSs vary significantly throughout the day with the changing SZA. For a meaningful comparison, a temporal constraint is therefore necessary. We have chosen the temporal difference be-
tween the measurements of the two FTSs to be less than 30 min, to restrict the difference in distance along the line-of-sight to approximately 50 km. Note that a stricter time constraint did not lead to a better comparison between the two instruments. In the case when more than one PARIS-IR observation matched the coincidence criterion (which happens regularly since PARIS-IR observations are taken every 7 min), the mean of all the coincident PARIS-IR measurements was used to compare with one Bruker 125HR retrieval.

As previously mentioned, the observed trace gas amounts can vary considerably depending on whether air masses are measured inside or outside the polar vortex. Thus, we have additionally included a criterion that restricts the difference of the sPV at 20 km along the line-of-sight of the two instruments, so that the maximum difference cannot exceed $0.3 \times 10^{-4}\,\mathrm{s}^{-1}$ (Batchelor et al., 2010). Note, this criterion is included as a precaution and a significant difference of the sPV for the two instruments does not occur frequently within the maximum temporal difference of 30 min.

As described in Sect. 2.2 and Sect. 2.3, the spectral resolution of the two FTSs is different. The instrument resolution can affect the retrieved total columns, since the retrieval from measurements with a higher resolution instrument is typically less influenced by the a priori profile and has larger DOFS (Rodgers, 2000). The retrievals from PARIS-IR spectra therefore typically result in fewer DOFS than the retrievals from the Bruker 125HR (see Table 1). The different resolutions are accounted for by smoothing the VMR profiles following the method described in Rodgers and Connor (2003). The improvement in the
intercomparsion of ground-based FTSs due to smoothing has been shown in numerous publications (e.g., Batchelor et al., 2010; Griffin et al., 2013) and is applied in this study. The smoothed profile $\boldsymbol{x}_{\mathrm{smooth}}$ is estimated by:

$$\boldsymbol{x}_{\mathrm{smooth}} = \boldsymbol{x}_{\mathrm{a}} + \mathbf{A} \cdot (\boldsymbol{x}_{\mathrm{h}} - \boldsymbol{x}_{\mathrm{a}}), \tag{1}$$

where the profile, $\boldsymbol{x}_{\mathrm{h}}$, was retrieved by the spectrometer having higher vertical resolution (Bruker 125HR), and is linearly interpolated onto the lower resolution instrument (PARIS-IR) retrieval grid and smoothed with the PARIS-IR averaging kernel,
$\mathbf{A}$, and a priori profile, $\boldsymbol{x}_{\mathrm{a}}$. The total or partial columns for these smoothed profiles have been calculated by integrating the smoothed VMR profile, $\boldsymbol{x}_{smooth}$, and the atmospheric density throughout the altitude range. This is consistent with the total column calculation method used within SFIT4.



## 3.2 Results and Discussion

The results of the comparisons of the total columns between PARIS-IR and the Bruker 125HR for all measurements satisfying the coincidence criteria (as defined above) are shown in Table 3 for the comparisons between 2007 and 2013. The total column differences were calculated as ([PARIS-Bruker]/[0.5×(PARIS+Bruker)]) for individual pairs and then averaged. Note that the

Bruker 125HR was installed in Eureka during the summer of 2006 and thus there are no coincident measurements for 2006. Figure 2 shows the correlation of the total column measurements from the two instruments during the campaigns between 2007 and 2013. The figure displays both smoothed (red dots) and unsmoothed (cyan triangles) total columns for the Bruker 125HR and the slopes of each regression plot (the regression analysis assumes errors on both axes). The black line is the regression plot for the smoothed total columns and the thin grey line is that for the unsmoothed total columns. The 1-to-1 correlation line

is included as a reference (black dashed line).

For most species ($O_3$, $HNO_3$, $CH_4$, and $N_2O$), as can be seen in Fig. 2, the differences between the Bruker 125HR smoothed and unsmoothed total columns are very small and typically less than 1 %, a negligible amount compared to the total retrieval uncertainty (see Table 1). The difference between the HCl, CO and $C_2H_6$ is some what larger, between 3 % and 4 %. The differences between the smoothed and unsmoothed columns are relatively large $\sim 9\,\%$ for HF, for which the total column

retrievals from PARIS-IR have DOFS of approximately 1, whereas the Bruker 125HR HF retrievals have twice as large DOFS (see Table 1). This suggests that the PARIS-IR columns are more influenced by the a priori than the Bruker 125HR retrievals. Thus, for this species, it is important to consider the different vertical resolutions of the retrieval from the two FTSs. Although the differences between the smoothed and unsmoothed Bruker 125HR retrievals are very small (less than 1 %) for $O_3$, $HNO_3$, $CH_4$, and $N_2O$, an approach has been taken that utilizes only the smoothed columns for the higher resolution instrument to

provide consistent analysis. Therefore, only differences between PARIS-IR and the smoothed Bruker 125HR total columns are discussed in the following.

The correlation is excellent for $O_3$, HCl, $HNO_3$, and CO, with correlation coefficients of $R \geq 0.95$ and the slopes of the regression plot between 0.93 and 1.13, suggesting that there is no significant bias. This is also apparent in the mean differences $\pm$ standard error, which are all small ($-0.33 \pm 0.10\,\%$ for $O_3$, $-2.37 \pm 0.13\,\%$ for HCl, $0.72 \pm 0.13\,\%$ for $HNO_3$,

and $4.56 \pm 0.09\,\%$ for CO) compared to the combined retrieval uncertainty (based on the total uncertainties of the retrieval from each instrument, see Table 1, that are added in quadrature). These combined retrieval uncertainties are $\pm 6.1\,\%$ for $O_3$, $\pm 3.1\,\%$ for HCl, $\pm 26.9\,\%$ for $HNO_3$, and $\pm 5.0\,\%$ for CO.

The comparison between the PARIS-IR and Bruker 125HR total columns is very good for $CH_4$ and $N_2O$, since the difference of $2.41 \pm 0.07\,\%$ for $CH_4$ is significantly smaller than the combined retrieval uncertainty ($\pm 10.5\,\%$, see Table 1) and the

difference of $-3.80 \pm 0.08\,\%$ for $N_2O$ is smaller than the combined retrieval uncertainty ($\pm 5.1\,\%$, see Table 2). The slopes of the regression plots for $CH_4$ and $N_2O$ are approximately 0.9 and 0.8, respectively. However, these PARIS-IR and Bruker 125HR total column retrievals are not very well correlated, with $R \sim 0.5$. This low correlation is likely due to the lack of variability observed compared to the retrieval uncertainty of the total columns of these gases, as the total columns only vary by approximately 10 % around $3.5 \times 10^{19}\,\mathrm{molec/cm^2}$ and $5.5 \times 10^{18}\,\mathrm{molec/cm^2}$ for $CH_4$ and $N_2O$, respectively. This variation is





within the combined total retrieval uncertainty for $CH_4$, and about half as much as the combined retrieval uncertainty of $N_2O$. It should be noted that the correlation is higher, $R > 0.85$ for $CH_4$ and $N_2O$, if the partial columns (using the same altitude range as for the PARIS-IR and ACE-FTS comparison in the following section, see Table 6) are considered for this comparison since the variation of the partial columns is higher than the total retrieval uncertainty. This is likely due to the fixed altitudes of the partial columns (see Table 6) and, as such, these partial columns can be influenced by subsidence inside the polar vortex.

The ground-based comparison between the $C_2H_6$ total columns is good, with a mean difference of $5.82 \pm 0.11\,\%$, which is smaller than the combined retrieval uncertainty ($\pm 6.7\,\%$). This agreement is also shown in the regression slope of $1.00 \pm 0.01$ and the high correlation of $R = 0.88$.

The PARIS-IR and Bruker 125HR HF total columns have a high correlation of $R = 0.89$. The slope of the regression plot $(0.63 \pm 0.02)$ suggests a negative bias between the HF datasets that can be seen in the mean difference of the total columns of $-7.68 \pm 0.27\,\%$. This bias is mainly apparent for large HF amounts with total columns greater than $3.0 \times 10^{15}\,\mathrm{molec/cm^2}$. The lower HF columns lie very close to the 1-to-1 correlation line. The total difference ($-7.68 \pm 0.27\,\%$) is larger than the combined retrieval uncertainty of the two datasets ($\pm 4.5\,\%$) and suggests that the PARIS-IR HF retrievals underestimate the amount of HF in the atmosphere. This negative bias of the PARIS-IR HF retrieval has been seen previously by Fu et al. (2011), who compared to another ground-based FTS in Eureka (Environment Canada Bomem DA8). The absorption lines of HF are quite narrow and the observation can be problematic with a ground-based instrument like PARIS-IR due to its limited spectral resolution. Generally, due to the limited DOFS of PARIS-IR's HF retrieval, the retrieved columns tend to be closer to the a priori ($\sim 1.6 \times 10^{15}\,\mathrm{molec/cm^2}$) and, therefore, issues arise in retrieving high HF amounts. Relaxing the covariance matrix constraint within the PARIS-IR retrieval resulted in oscillations of the retrieved HF profile and was not able to resolve this issue.

Following this discussion of the mean differences for the entire dataset between 2007 and 2013, next we focus on individual years during this time period. Little variation of the differences was found and they were within the combined retrieval uncertainty in most years for most species (except for HF). These yearly mean differences of the smoothed total columns together with the standard error can be found in Figs. 3 and 4 for the stratospheric and tropospheric species, respectively (yellow bars, for the PARIS-IR and Bruker 125HR total column comparison). The number of pairs compared varies for each year, and is displayed above or below the bars in the figure. The difference in numbers of coincident pairs is mainly due to the different weather conditions for each year. For example in 2009 and 2010, there were many days of sunshine and little to no cloud cover, and measurements could be taken almost every day throughout the campaign. There is interannual variation of the retrieval differences between PARIS-IR and the Bruker 125HR, however, these are within the combined retrieval uncertainties for most species during most years. For the HCl comparison, two years (2010 and 2013) are outside the combined retrieval uncertainty. For the comparisons of $N_2O$, CO, and $C_2H_6$, one year in each case is outside the combined retrieval uncertainty. Overall, no significant degradation of the comparison could be found over the seven year period for any of the eight retrieved species.

To conclude, we found that after accounting for the different resolutions by smoothing, the mean differences between PARIS-IR and the Bruker 125HR total columns are below $4\,\%$ and within the estimated combined retrieval uncertainties for all species, with the exception of HF. These differences and correlation coefficients are comparable or slightly better for some species





compared to previous side-by-side instrument comparisons for PARIS-IR (that used SFIT2) in Eureka (e.g., Batchelor et al., 2010; Fu et al., 2011) and at other locations in North America (e.g., Wunch et al., 2007; Griffin et al., 2013).

# 4 Comparison between ACE-FTS and the ground-based FTSs

## 4.1 Methodology

As described in the introduction, instrument comparisons between ground- and satellite-based remote sensing instruments in the high Arctic, especially during the springtime when the Sun rises, are challenging, primarily due to the polar vortex. Here we apply a comparison method similar to Batchelor et al. (2010) for the stratospheric species ($O_3$, $HCl$, $HNO_3$, and $HF$) that are quite different inside to outside the polar vortex. Measurements are considered coincident if they were recorded within 12 h of each other and when the distance at specific altitudes along the line-of-sight (between 14 km and 40 km) is less than 1000 km. Additionally, for those same altitudes, the difference in sPV between the measurements was restricted to a maximum of $0.3 \times 10^{-4}\,\mathrm{s}^{-1}$ and the difference of the temperature was limited to less than 10 K. This ensures that similar air masses are observed by both instruments. While these criteria filter dissimilar air masses in most cases, this may still allow occasionally for dissimilar air masses to be compared right near the edge of the polar vortex.

For the retrievals of the tropospheric trace gases from the ground-based FTSs ($CH_4$, $N_2O$, $CO$, and $C_2H_6$), an effort was made to compare partial columns with a lower boundary as far into the troposphere as possible. This lowers the number of ACE-FTS occultations that can be compared, but improves the comparison. As for the stratospheric species comparison, measurements are considered coincident if they were recorded within 12 h of each other. However, to have enough observations to compare, a less strict criterion has been used, where only the distance along the line-of-sight is considered at 14 km. Including the difference in sPV criterion along the line-of-sight at 14 km did not impact the comparison, since none of the differences were found outside this criterion. Furthermore, we varied the distance criterion between a maximum of 500 km and 1000 km along the line-of-sight at 14 km. As will be discussed in detail in the following section, the tighter distance criterion improves the correlation between the partial columns significantly for the tropospheric species. A distance of 1000 km seems too large to compare some tropospheric species that can vary considerably based on location. For $CH_4$ and $N_2O$ the total column variability is low (as discussed in Sect. 3.2), however, we observed that partial columns show significantly more variation. This is consistent with many other validation papers which have used 500 km as a limit for coincident measurements for tropospheric species in the high Arctic (e.g., Strong et al., 2008; Viatte et al., 2014).

In order to compare the space-borne to the ground-based FTS measurements, partial columns have to be considered. This is because ACE-FTS measurements are not made in the lower troposphere. The partial column altitude ranges are specific to each species and are based on where the ground-based instrument retrievals are the most sensitive (see Sect. 2.4 and Fig. 1) and on the observation lower altitude limit of ACE-FTS. The altitude ranges of the partial columns are slightly different for each ground-based FTS, since the instruments' resolutions, and therefore the sensitivities, are not the same. These ranges for each partial column are listed in Table 4 to Table 7, for each instrument and species.





Smoothing has been applied to ACE-FTS profiles with either the PARIS-IR or the Bruker 125HR averaging kernels in a similar manner as described in Sect. 3.1. ACE-FTS profiles have been interpolated from its 1-km altitude grid to the 29- and 47-layer altitude grids used for the PARIS-IR and Bruker 125HR retrievals, respectively. In the lower troposphere, where no ACE-FTS measurements are available, the ground-based a priori values are used for the calculation. These interpolated ACE-FTS

VMR profiles are then smoothed with the averaging kernel and the a priori of the comparison instrument, as described in Eq. 1. The partial columns are then calculated from the smoothed profiles, based on the partial column altitude ranges (see Tables 4-7). The included a priori values, in the lower troposphere, are not considered in this partial column calculation. This method is consistent with other validation studies that have compared satellite-based instruments to ground-based FTSs (e.g., Vigouroux et al., 2007; Kerzenmacher et al., 2008; Batchelor et al., 2010). The partial column differences were calculated as ([GB-

ACE]/[$0.5\times$(GB+ACE)]), where GB is the ground-based instrument, either PARIS-IR or the Bruker 125HR as applicable. If more than one ground-based measurement is coincident with a particular ACE-FTS occultation, the mean of all coincident ground-based measurements was used to calculate the difference.

## 4.2   Results and Discussion

The mean partial column differences for the stratospheric species from 2006 to 2013 between ACE-FTS and PARIS-IR are

shown in Table 4, and between ACE-FTS and the Bruker 125HR (from 2007 to 2013) in Table 5. For each of the stratospheric species, approximately 120 and 100 satellite occultations were found to be coincident with PARIS-IR and the Bruker 125HR measurements, respectively. The smaller number of coincident measurements with the Bruker 125HR is partly due to the shorter time period (2007-2013) for the comparisons, but also because individual species are not measured as often as with PARIS-IR. The number of coincident measurements varies annually and due to a data processing gap for ACE-FTS in 2012,

no coincident occultations were found that year.

Very good agreement was found between ACE-FTS and both ground-based FTSs for $O_3$ and HCl. The correlations for these gases are excellent, with the coefficient of correlation $R \geq 0.91$ and the slopes of the regression plot are close to the 1-to-1 line ($> 0.81$; see Table 4 and Table 5). The mean differences $\pm$ standard error for the comparison to both ground-based FTSs for the $O_3$ partial columns ($3.5\pm0.6\,\%$ for PARIS-IR and $3.6\pm0.6\,\%$ for the Bruker 125HR) are within the total retrieval uncertainties

of $\pm3.5\,\%$ (PARIS-IR) and $\pm5.0\,\%$ (Bruker 125HR), respectively (see Table 1). The mean differences for the HCl comparisons ($-1.0\pm0.6\,\%$ for PARIS-IR and $2.4\pm0.6\,\%$ for the Bruker 125HR) are within the total retrieval uncertainty of $\pm2.5\,\%$ from PARIS-IR and are 0.5\,\% larger than the total retrieval uncertainty of $\pm1.9\,\%$ for the Bruker 125HR.

The ground-based $HNO_3$ partial columns are in good agreement with ACE-FTS, for which the mean differences of $5.6\pm0.8\,\%$ between PARIS-IR and ACE-FTS and $1.5\pm1.0\,\%$ between the Bruker 125HR and ACE-FTS, respectively, are negligible

compared to the very large total retrieval uncertainty of $\pm19\,\%$. The correlation between ACE-FTS and the ground-based partial columns is high ($R \geq 0.77$) and the slope of the regression plot is greater than 0.77.

The ACE-FTS HF partial columns agree well with the Bruker 125HR, for which the mean difference ($-1.9\pm1.0\,\%$) is approximately half of the Bruker 125HR total retrieval uncertainty of $\pm3.5\,\%$. The correlation of those partial columns is high with $R = 0.84$ and the slope of regression of 0.91. The comparison between ACE-FTS and PARIS-IR HF partial columns



is not as good, since the mean difference ($-6.1 \pm 1.2\,\%$) is more than twice PARIS-IR's total retrieval uncertainty ($\pm 2.5\,\%$). Large differences are mainly observed when high HF concentrations are measured by ACE-FTS. And, although the slope of the regression plot is close to the 1-to-1 line, the correlation between the partial columns is relatively low ($R = 0.59$). This negative difference of the PARIS-IR HF retrieval, especially for high HF columns, is consistent with the bias found from

comparison of the total columns to the Bruker 125HR, see Sect. 3.2.

Next we consider the variation of the mean differences for each individual year between 2006 and 2013. The yearly differences are displayed in Figs. 3 and 4 for the stratospheric and tropospheric species, respectively: The ACE-FTS and PARIS-IR comparisons are shown as blue bars and the ACE-FTS and Bruker 125HR comparisons are displayed as cyan bars. Also, displayed are the partial column comparisons between PARIS-IR and the Bruker 125HR (red bars) to better understand the

impacts of comparing partial columns rather than total columns. The largest difference between using partial columns and total columns can be seen for HF, where the partial column differences are approximately twice as large compared to the total column differences. This is due to the small DOFS of the PARIS-IR HF retrieval, for which the partial columns have generally less than 1 DOFS ($\sim 0.8$). The impact of the total column versus the partial column comparison is not as significant for $O_3$, HCl, and $HNO_3$.

Also shown in Figs. 3 and 4 are the standard errors of the yearly mean differences (shown as error bars), and the number of pairs (e. g. number of ACE-FTS occultations) used to estimate these differences (listed below or above the bars). Variation of the annual mean differences are apparent for the ground-based versus ACE-FTS comparisons. Generally, for all four stratospheric species, the absolute mean differences do not appear to increase between 2006 and 2013. As expected, it was found that whether measurements were taken inside or outside the polar vortex has a significant impact on the comparison, especially for $O_3$. The

yearly average and standard deviation of the sPV along the line-of-sight at $20\,km$ is displayed in the lower panel of Fig. 3, together with the inner and outer polar vortex edge as a reference. In 2009 and 2013, on average measurements that are used for the comparison were taken at the edge of the polar vortex. For measurements in those years, the $O_3$ and $HNO_3$ comparisons seem to result in larger differences. In 2007 and 2011, all measurements that are used for the comparison were taken inside the polar vortex, where on average ACE-FTS's $O_3$ partial columns were larger than the ground-based ones. The reverse was

seen in all other years when the comparison was made primarily outside the polar vortex. In 2011, the mean difference for HCl between ACE-FTS and the Bruker 125HR seems to be very large, however, only two occultations were compared to the ground-based measurements. Furthermore, the partial columns from both instruments were approximately four times smaller than in previous years ($1.3 \times 10^{15}\,molec/cm^2$ ), which impacts the percentage difference; the absolute difference of those columns was approximately $0.2 \times 10^{15}\,molec/cm^2$ , which is comparable with other years. The comparison of HF between

ACE-FTS and PARIS-IR seems to be worse in 2007 and 2011, compared to previous years. The PARIS-IR HF partial columns were larger ($\sim 2.3 \times 10^{15}\,molec/cm^2$) than in other years ($\sim 1.8 \times 10^{15}\,molec/cm^2$) and tended to be closer to the a priori ($\sim 1.6 \times 10^{15}\,molec/cm^2$), see Sect. 3.2.

As mentioned in Sect. 4.1, the tropospheric species from the satellite- and ground-based measurements required different criteria than were used for the comparison of the stratospheric species. The sPV and temperature profile criteria that were

applied for the previous comparison have been removed in order to have sufficient observations available for the comparison.



The impact of the distance criteria was investigated using the maximum distance between the instruments (at $14\,km$ along the line-of-sight). Table 6 and Table 7 summarize the ACE-FTS and PARIS-IR, and ACE-FTS and Bruker 125HR comparisons, respectively, for coincident measurements at a maximum distance of $1000\,km$ and of $500\,km$. A maximum distance of $500\,km$ shows significantly improved correlation compared to using $1000\,km$ between the satellite- and ground-based instruments'

partial columns. For this, the correlation coefficient $R$ increases on average from $0.6$ to $0.8$. However, this distance criterion has a relatively small impact on the mean differences for $CH_4$ and $N_2O$. As such, the mean differences for those species using a stricter distance criterion are within the standard error of the differences found with a $1000\,km$ distance criterion. Since the mean differences are quite similar for the two different distance criteria, but the correlation is significantly improved, only the mean differences, correlation coefficients and correlation slopes at a maximum distance of $500\,km$ are considered in the

following discussion. Tightening the distance criteria any further results in very few measurements that are selected for the comparisons. It should also be noted that the different altitude ranges (listed in Table 6 and Table 7) selected for the partial column comparison between ACE-FTS and PARIS-IR, and ACE-FTS and the Bruker 125HR contribute to differences in the number of pairs that are compared (approximately one third less for the Bruker 125HR).

     The $CH_4$ partial column datasets agree well. The mean differences between ACE-FTS and PARIS-IR ($3.0 \pm 0.3\,\%$), and

ACE-FTS and Bruker 125HR ($0.6 \pm 0.4\,\%$) partial columns are well within the estimated total retrieval uncertainty of the ground-based instruments ($\pm 6.8\,\%$ for PARIS-IR and $\pm 8.0\,\%$ for the Bruker 125HR). There is a high correlation between the instruments' partial columns with $R = 0.78$ and the slope of the regression plot of $0.97$ for the PARIS-IR. The correlation is also high for the Bruker 125HR comparison with $R = 0.89$ and the slope of the regression plot around $0.79$.

     The $N_2O$ partial columns of ACE-FTS agree well with those of the Bruker 125HR, for which the mean difference ($-1.6 \pm$

$0.5\,\%$) is approximately half of its total retrieval uncertainty ($\pm 3.7\,\%$), with a high correlation between those partial columns ($R = 0.84$) and a slope of the regression plot of $0.73$. The mean difference for the ACE-FTS and PARIS-IR comparison is larger ($6.6 \pm 0.5\,\%$) than the estimated total retrieval uncertainty of PARIS-IR ($\pm 3.5\,\%$), however, the correlation between the partial columns is high ($R = 0.8$ with a slope of the regression plot of $1.05$).

     ACE-FTS partial columns of CO and $C_2H_6$ are only compared to the Bruker 125HR. The PARIS-IR partial columns of

those species have less than 0.5 DOFS ($\sim 0.3$), which is not ideal for an instrument comparison (Vigouroux et al., 2007). The correlation of the partial columns for both species is quite high ($R > 0.75$) and the slope of the regression plot is close to the 1-to-1 line ($\geq 0.75$). However, the mean differences are quite large $7.1\,\%$ for CO and $20\,\%$ for $C_2H_6$, respectively. Note, that the mean difference for CO is more than twice as large ($16.7 \pm 3.3\,\%$) in 2010 compared to all other years. Excluding 2010 from the comparison leads to a mean difference of $3.28\,\%$ that is within the total retrieval uncertainty ($\pm 3.5\,\%$). In 2010, a

number of slightly enhanced CO columns were observed by both ground-based FTS instruments near Eureka that were not observed by the ACE-FTS and could be a local enhancement.

     Looking at the annual variability of the instruments' mean differences (see Fig. 4), relatively small year-to-year variability, that is within the ground-based total retrieval uncertainty, can be seen for the $CH_4$ and $N_2O$ partial column comparisons. As noted above, the CO difference in 2010 is significantly larger than in all other years, likely due to a localized enhancement.

The $C_2H_6$ annual partial column differences vary between $6\,\%$ and $34\,\%$.



To conclude, very little bias was seen between ACE-FTS and both ground-based FTSs for the comparison of the stratospheric species (except for the comparison to PARIS-IR HF). There is a negative bias for the comparison between the HF partial columns from PARIS-IR and ACE-FTS, which is consistent with the bias seen in the ground-based comparisons (see Sect. 3.2). The differences between ACE-FTS and the Bruker 125HR for $O_3$, HCl, $HNO_3$, and HF between 2006 and 2013 using SFIT4

and ACE-FTS v3.5, are consistent with Batchelor et al. (2010) for 2007 and 2008 using SFIT2 and ACE-FTS v2.2+updates. For stratospheric species, the distance criterion of $1000\,\mathrm{km}$ is sufficient, however, the comparison for tropospheric species is improved if the distance is limited to $500\,\mathrm{km}$. ACE-FTS v3.5 $CH_4$, $N_2O$, and CO partial columns compare well to the ground-based retrievals. The mean differences found for the tropospheric species (with ACE-FTS v2.2+updates) are comparable with Strong et al. (2008) and De Mazière et al. (2008) for $N_2O$ and $CH_4$, respectively; and are improved by more than $15\,\%$ for CO

compared to Clerbaux et al. (2008). This improvement could be due to the latest retrieval version of ACE-FTS (v3.5) and also to the ground-based retrieval algorithm (SFIT4) that has been used for this study. We found that with the new retrieval algorithm SFIT4 and latest NDACC/IRWG recommendations CO has a higher sensitivity in the lower stratosphere compared to previous retrievals. Furthermore, it was shown that the comparison between the two ground-based instruments did not degrade over this time period. The mean differences change slightly each year for all species, but did not increase over time.

## 15 5 Evolution of the stratospheric and tropospheric species during Arctic springtime, 2006-2013

The dataset displayed in Fig. 5 shows the springtime campaign average, late February to early April, (and standard deviation) obtained from the PARIS-IR dataset for each year. This dataset consists of yearly springtime average total column measurements between 2006 and 2013 for the eight species, as well as the yearly springtime average sPV at $20\,\mathrm{km}$ along the line-of-sight. The red solid lines represent the line of best fit (first order polynomial fit) and the black dashed lines display

the standard deviation of the fit. The outer and inner edges of the polar vortex are marked in the lower panel by cyan dashed and solid lines, respectively. The yearly variation of the stratospheric species ($O_3$, HCl, $HNO_3$, and HF) is highly influenced by the dynamics of the stratosphere and the strength of the polar vortex. Note, that the dataset shown in Fig. 5 has not been filtered for observations taken inside or outside of the polar vortex. What is immediately apparent is that $O_3$, HCl and $HNO_3$ columns are very low in 2011. In this year, measurements were mainly sampled inside a strong polar vortex, as can be seen

in the lower panel of Fig. 5 as well as in Lindenmaier et al. (2012). The vortex remained near Eureka for the whole month of March, so ground-based observations were mainly taken inside the vortex (see Fig. 5 (f) in Lindenmaier et al., 2012). A strong and cold vortex is typically associated with chemical $O_3$ depletion and denitrification (WMO, 2014), and we see the averages for $O_3$, HCl, $HNO_3$ are low in 2007 and significantly lower in 2011 than in all other years. The location of air sampled by the instruments with respect to the polar vortex has a high interannual variability over Eureka between 2006 and 2013. Figure

6 shows the averages of the stratospheric species outside the polar vortex, when the sPV is less than $1.2 \times 10^{-4}\,\mathrm{s}^{-1}$ at $20\,\mathrm{km}$ along the line-of-sight (see Sect. 2.6). The polar vortex near Eureka was not strong until the middle of March in 2007, while it remained strong until the end of March in 2011. Therefore, the airmass outside the polar vortex was seen for only for a few days in early March in 2007, and in late March to early April in 2011. Thus, there is the potential for a bias due to when





measurements were made outside the polar vortex in 2007 and 2011. The dataset inside the polar vortex is not shown here, due to the strong interannual variability of the polar vortex and the chlorine activation processes.

In examining PARIS-IR's eight-year dataset, we can estimate whether the changes seen in the dataset over this time period are statistically significant. To determine whether or not it is possible to assess a trend, a number of factors need to be considered:

the time period of the dataset, the magnitude of the trend $w_o$, the variability $\sigma$ and the autocorrelation $\phi$ of the noise of the dataset. This is described in detail in Weatherhead et al. (1998). The minimum number of years, $n^*$, needed to observe a trend, can be estimated by:

$$n^* = \left[ \frac{3.3 \cdot \sigma}{|w_o|} \cdot \sqrt{\frac{1+\phi}{1-\phi}} \right]^{2/3}. \tag{2}$$

Using the slopes of the lines of best fit of Fig. 5 and 6 for each species, we can determine whether or not a trend can be detected

in our dataset based on the number of years compared to the estimated minimum number of years $n^*$, computed from Eq. 2.

The total columns of all stratospheric species sampled outside the polar vortex in the springtime (Fig. 6) show an increase between 2006 and 2013. The lines of best fit $\pm$ standard deviation indicate an increase of $0.9 \pm 1.2\,\%\mathrm{yr}^{-1}$ for $O_3$, $1.7 \pm 0.8\,\%\mathrm{yr}^{-1}$ for HCl, $1.7 \pm 0.7\,\%\mathrm{yr}^{-1}$ for $HNO_3$, and $3.8 \pm 1.4\,\%\mathrm{yr}^{-1}$ for HF, respectively. Using the method above, the minimum number of years required to detect a trend of these magnitudes from this dataset is approximately 5 years for $O_3$, 6

years for HCl, 7 years for HF, and 9 years for $HNO_3$. With this eight-year dataset, trends are likely detected in HCl and HF in the atmosphere of the high Arctic (outside the polar vortex) from PARIS-IR measurements. Although it seems there are enough years available to detect a trend in $O_3$, it should be noted that the uncertainty of the increase seen in $O_3$ is larger than the actual estimated increase. However, recent increasing stratospheric $O_3$ (in the tropics and mid-latitudes) has been previously reported by Harris et al. (2015) using satellite and ozonesonde observations, and our findings for high latitudes are consistent with their

results. The magnitude of the increase of HCl at northern high latitudes is consistent with Mahieu et al. (2014), and is assumed to be due to atmospheric circulation changes in the northern hemisphere. These changes occurred after 2005/2006 and are possibly on a short time-scale (Mahieu et al., 2014). The increase of HF is likely due to the increase of $COF_2$ that has been discussed in Harrison et al. (2014). A longer dataset is necessary to be able to observe a trend for $HNO_3$.

For the tropospheric species, no sPV filter has been applied, since the influence of the polar vortex is not as significant

in the troposphere. Looking at Fig. 5, it seems that $CH_4$ and $C_2H_6$ are increasing each year since 2006. Between 2006 and 2013, the $CH_4$ columns increased by approximately $0.5 \pm 0.1\,\%\mathrm{yr}^{-1}$, and the $C_2H_6$ columns increased by approximately $1.6 \pm 0.2\,\%\mathrm{yr}^{-1}$. However, $C_2H_6$ has started to increase at a higher rate since 2009 as can be seen from Fig. 5. Between 2009 and 2013, $C_2H_6$ increased by $2.3 \pm 0.5\,\%\mathrm{yr}^{-1}$ based on our dataset. CO appears to be decreasing slightly over the time period between 2006 and 2013, by approximately $-0.8 \pm 0.6\,\%\mathrm{yr}^{-1}$. $N_2O$ seems relatively constant in the Arctic spring

between 2006 and 2013 and the slight increase seen over this time period is well within the standard deviation with a slope of $0.3 \pm 0.3\,\%\mathrm{yr}^{-1}$. The minimum number of years to detect a trend from these datasets, based on Weatherhead et al. (1998), are: 8 years for $CH_4$ and $C_2H_6$, 9 years for CO, and more than 10 years for $N_2O$. Based on the available data and Eq. 2, we can detect a trend for $CH_4$ in the high Arctic between 2006 and 2013. For $C_2H_6$, CO, and $N_2O$ our dataset is not long enough to observe a trend. Increasing $C_2H_6$ starting in 2009 has also been reported in previous studies (Franco et al., 2015;



Franco et al., 2016). In the Arctic, Franco et al. (2016) have found increasing $C_2H_6$ of approximately $3 \pm 1\,\%\mathrm{yr}^{-1}$ near Eureka and Thule, Greenland between 2009 and 2014. A decrease of CO has been reported above the high Arctic station in Kiruna by ground-based FTS observations of $-0.61 \pm 0.16\,\%\mathrm{yr}^{-1}$ (Angelbratt et al., 2011). These results are consistent with our observations. For $CH_4$, no recent changes in the Arctic have been reported yet, however, at lower latitudes increasing $CH_4$ has

been found. Sussmann et al. (2012) observed a $0.3\,\%\mathrm{yr}^{-1}$ increase of $CH_4$ between 2007 and 2011 in Garmisch and Zugspitze, Germany from ground-based FTS measurements. This is similar to the $CH_4$ increase that we found in the high Arctic near Eureka ($0.5\,\%\mathrm{yr}^{-1}$, 2006-2013).

To conclude, we have found that with our dataset we can detect a trend for HCl, HF, $O_3$, and $CH_4$ near Eureka between 2006 and 2013. Total columns of all these species are increasing over this time period.

## 6  Summary and conclusions

We have presented eight-years of measurements between 2006 and 2013 in the high Arctic, with the purpose of providing a detailed comparison between the two ground-based FTS instruments and the space-borne ACE-FTS as well as examining atmospheric composition change over this period. In total, eight atmospheric gases have been utilized and assessed, namely $O_3$, HCl, $HNO_3$, HF, $CH_4$, $N_2O$, CO, and $C_2H_6$.

Side-by-side instrument comparisons were carried out for the two ground-based FTSs at PEARL during the Canadian Arctic ACE Validation Campaigns, from 2007 to 2013. With respect to the smoothed total columns, the instrumental differences are well within the estimated combined retrieval uncertainties and below $6\,\%$ for most species (except HF) and the retrieved columns are highly correlated ($R > 0.85$) for the two FTSs. Our results are comparable with ground-based side-by-side comparisons with PARIS-IR, such as those reported by Batchelor et al. (2010) and Griffin et al. (2013). Smoothing the retrieved

profiles of the Bruker 125HR with PARIS-IR's averaging kernel provides a more accurate comparison and has been done for all species. However, this is only significant for gases with low DOFS such as $C_2H_6$ and HF. The comparison also showed that HF total columns are slightly underestimated by PARIS-IR versus the Bruker 125HR HF columns. Overall, these comparisons contribute to the satellite validation effort for ACE-FTS in the high Arctic with the latest retrieval algorithm SFIT4. It was further found that the comparisons did not degrade during this time period.

The partial column comparisons between ACE-FTS v3.5 and the two ground-based FTSs were carried out over this eight-year period. For $O_3$, HCl, $HNO_3$, and HF coincidence criteria including sPV and temperatures along the line-of-sight were employed. The resulting mean biases are smaller than $4\,\%$ and within the estimated uncertainty of the ground-based retrieval for all species for the Bruker 125HR comparison. The mean bias between ACE-FTS and PARIS-IR was within approximately $6\,\%$ for all species. Our results have shown that the correlation between the datasets is significantly improved (by $\sim 0.2$)) when

the maximum distance is limited to $500\,\mathrm{km}$ for the comparisons of tropospheric species. For $CH_4$, $N_2O$, and CO, the biases are smaller than $3.5\,\%$ and less than the ground-based total retrieval uncertainty for the comparison between ACE-FTS and the Bruker 125HR. The PARIS-IR $CH_4$ and $N_2O$ columns agree well with ACE-FTS, with differences of $3.0\,\%$ and $6.6\,\%$, respectively. The mean differences of the ACE-FTS and the Bruker 125HR $C_2H_6$ partial columns are $\sim 20\,\%$; however, a high



correlation ($R = 0.75$) between these datasets was found. Overall, the results show that ACE-FTS 2006-2013 retrievals are consistent with ground-based observations, even in 2013, a decade after the instrument was launched. No increasing mean differences of the yearly comparisons were found over this time period. The long-term ground-based FTS measurements continue to contribute to the validation of the trace gas amounts retrieved from measurements from the ACE-FTS instrument

on-board SCISAT.

During this entire time period (2006-2013), increasing $O_3$ ($0.9\,\%\mathrm{yr}^{-1}$), HCl ($1.7\,\%\mathrm{yr}^{-1}$), HF ($3.8\,\%\mathrm{yr}^{-1}$), $CH_4$ ($0.5\,\%\mathrm{yr}^{-1}$) and $C_2H_6$ ($2.3\,\%\mathrm{yr}^{-1}$, 2009-2013) have been found near Eureka in the springtime. These results were compared to previously published measurements from different datasets and at different locations. Overall, our estimated increases are consistent with the values reported by Harris et al. (2015) for $O_3$, Mahieu et al. (2014) for HCl, and Sussmann et al. (2011) for $CH_4$, respec-

tively. As such, our results from the ground-based PARIS-IR dataset complement their findings by showing that these increases are also apparent in the high Arctic.

*Acknowledgements.* The Canadian Arctic ACE Validation Campaigns are funded by the Canadian Space Agency (CSA), Environment and Climate Change Canada (ECCC), the Natural Sciences and Engineering Research Council of Canada (NSERC), and the Northern Scientific Training Program. CANDAC and PEARL are supported by the Atlantic Innovation Fund/Nova Scotia Research Innovation Trust,

Canadian Foundation for Climate and Atmospheric Sciences, Canada Foundation for Innovation, CSA, ECCC, Government of Canada International Polar Year funding, NSERC, Ontario Innovation Trust, Polar Continental Shelf Program and the Ontario Research Fund. The Atmospheric Chemistry Experiment (ACE) is mainly supported by the CSA and NSERC. We would like to acknowledge the ACE campaign and CANDAC operators: Ashley Harrett, Alexei Khmel, Paul Loewen, Oleg Mikhailov and Matt Okraszewski; as well as Keeyoon Sung, Emily McCullough, and Joseph Mendonca for maintaining and operating the ground-based FTSs. The authors would like to thank the staff

at the Eureka Weather Station and CANDAC for the logistical, on-site support provided at Eureka, and the launch of many radio- and ozone-sonde balloons for us. We are also very grateful to William Daffer from JPL, who carried out the DMP calculations.



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



**Table 1.** Summary of the parameters used for the eight trace gas retrievals (retrieval microwindows and interfering species) with an estimate of the total uncertainty of the retrieval (± in %), DOFS, and the RMS/DOFS values (in %) used to filter the dataset of the PARIS-IR and the Bruker 125HR retrievals. In each retrieval, a single or multiple microwindows are fitted simultaneously as listed in the table below. For the calculation of the total uncertainty and contributions, see the description given in the text.

| Gas | Microwindows (cm$^{-1}$) | Interfering Species | Total uncertainty (%) | | DOFS | | RMS/DOFS (%) | |
|---|---|---|---|---|---|---|---|---|
| | | | PARIS-IR | Bruker | PARIS-IR | Bruker | PARIS-IR | Bruker |
| $O_3$ | 1000.00–1004.50 | $\{$ $H_2O$, $CO_2$, $C_2H_4$, $O_3$ isotopologues | 3.5 | 5.0 | 2.5 | 4.5 | 6.0 | 3.5 |
| HCl | 2775.70–2775.80 2925.80–2926.00 | $O_3$, $N_2O$ $CH_4$, $NO_2$, $O_3$ | 2.5 | 1.9 | 1.0 | 2.5 | 2.5 | 0.6 |
| $HNO_3$ | 867.50–870.00 | $H_2O$, OCS, $NH_3$ | 19.0 | 19.0 | 1.5 | 2.5 | 4.0 | 2.2 |
| HF | 4038.81–4039.07 4109.77–4110.07 | $\{$ $H_2O$, $CH_4$, HDO | 2.9 | 3.5 | 1.0 | 2.0 | 4.0 | 0.75 |
| $CH_4$ | 2613.70–2615.40 2650.60–2651.30 2903.60–2904.03 2921.00–2921.60 2835.50–2835.80 | $\{$ HDO, $CO_2$, $NO_2$ HDO | 6.8 | 8.0 | 2.0 | 2.0 | 2.0 | 0.35 |
| CO | 2057.70–2058.00 2069.56–2069.76 2157.50–2159.15 | $\{$ $CO_2$, $O_3$, OCS $\{$ $CO_2$, $O_3$, OCS, $N_2O$, $H_2O$ | 3.5 | 3.5 | 1.5 | 3.5 | 3.0 | 1.2 |
| $N_2O$ | 2481.30–2482.60 2526.40–2528.20 2537.85–2538.80 2540.10–2540.70 | $\{$ $H_2O$, HDO, $CO_2$, $O_3$, $CH_4$ | 3.5 | 3.7 | 2.0 | 3.0 | 4.0 | 3.7 |
| $C_2H_6$ | 2976.66–2976.95 2983.20–2983.55 2986.50–2986.95 | $\{$ $H_2O$, $O_3$ | 5.0 | 4.4 | 1.0 | 2.0 | 3.0 | 1.5 |





**Table 2.** Error budget used to estimate the total retrieval uncertainties. The line width error (discussed in the text) is the combined uncertainty of the pressure and temperature broadening. The same uncertainties have been used to estimate the errors for PARIS-IR and the Bruker 125HR, with the exception of the SZA that is dependent on the measurement duration for each FTS. Details can be found in the text.

| Species | $O_3$ | HCl | $HNO_3$ | HF | $CH_4$ | CO | $N_2O$ | $C_2H_6$ |
|---|---|---|---|---|---|---|---|---|
| | \multicolumn Systematic error (fractional value) | | | | | | | |
| Line intensity | 0.05 | 0.015 | 0.2 | 0.035 | 0.075 | 0.035 | 0.035 | 0.04 |
| Pressure broadening | 0.035 | 0.015 | 0.075 | 0.015 | 0.075 | 0.015 | 0.035 | 0.04 |
| Temperature broadening | 0.075 | 0.15 | 0.2 | 0.015 | 0.15 | 0.035 | 0.075 | 0.04 |
| Temperature uncertainty | between 0.49 K and 1.44 K depending on altitude | | | | | | | |
| | Random error | | | | | | | |
| Temperature uncertainty | between 9.0 K and 0.63 K depending on altitude | | | | | | | |
| SZA uncertainty from change over measurement | $0.075°$ (PARIS-IR) and $0.06°$ (Bruker 125HR) | | | | | | | |

**Table 3.** Comparison of PARIS-IR and the smoothed Bruker 125HR total columns for all trace gases in this study, averaged between 2007 and 2013. $N$ is the number of coincident pairs involved in this calculation. The third column (TC diff) represents the mean differences between the total columns of the two FTSs (in %) along with the $1\sigma$ standard deviation and the standard error ($1\sigma/\sqrt{N}$; in brackets). The correlation coefficient ($R$) and the slope of the regression plot (slope), along with the uncertainty of the slope, are shown in columns 4 and 5, respectively.

| Species | $N$ | TC diff (%) | $R$ | slope |
|---|---|---|---|---|
| $O_3$ | 924 | $-0.33 \pm 3.12$ (0.10) | 0.98 | $0.92 \pm 0.01$ |
| HCl | 907 | $-2.37 \pm 3.84$ (0.13) | 0.96 | $1.04 \pm 0.01$ |
| $HNO_3$ | 1623 | $0.72 \pm 5.04$ (0.13) | 0.95 | $0.98 \pm 0.01$ |
| HF | 685 | $-7.68 \pm 7.03$ (0.27) | 0.89 | $1.04 \pm 0.02$ |
| $CH_4$ | 1055 | $2.41 \pm 2.24$ (0.07) | 0.48 | $0.91 \pm 0.02$ |
| $N_2O$ | 947 | $3.80 \pm 2.41$ (0.08) | 0.52 | $0.77 \pm 0.02$ |
| CO | 792 | $4.56 \pm 2.42$ (0.09) | 0.95 | $1.13 \pm 0.01$ |
| $C_2H_6$ | 1538 | $5.82 \pm 4.60$ (0.11) | 0.88 | $1.00 \pm 0.01$ |





**Table 4.** Comparison of ACE-FTS v3.5 and PARIS-IR partial columns (2006-2013) for the stratospheric gases presented in this study. $N$ is the number of coincident pairs used in this calculation. The third column gives the altitude range of the partial columns used for this comparison, the third column shows the mean distance between the observed air masses from the instruments along the line-of-sight at 20 km. The mean time between the observations is displayed in the fourth column and the mean beta angle of the ACE observations in the fifth column. The seventh column (PC diff) represents the mean differences in the partial columns between the FTSs (in %) along with the standard error. The correlation coefficient ($R$) and the slope of the regression plot (slope), along with the $1\sigma$ uncertainty of the slope, are shown in column 9.

| Species | $N$ | Altitude range (km) | Mean distance at 20 km (km) | Mean time difference (h) | Mean beta angle of ACE occultations (°) | PC diff (%) | $R$ | slope |
|---|---|---|---|---|---|---|---|---|
| $O_3$ | 118 | 9.5–51.0 | 608 | 3.7 | 26.5 | $3.5 \pm 0.6$ | 0.92 | $0.81 \pm 0.03$ |
| HCl | 117 | 9.5–41.5 | 601 | 3.6 | 26.9 | $-1.0 \pm 0.6$ | 0.95 | $0.98 \pm 0.03$ |
| $HNO_3$ | 119 | 9.5–33.5 | 605 | 3.7 | 26.6 | $5.7 \pm 0.8$ | 0.81 | $0.78 \pm 0.04$ |
| HF | 120 | 14.5–41.5 | 606 | 3.9 | 24.2 | $-6.1 \pm 1.2$ | 0.59 | $1.02 \pm 0.08$ |

**Table 5.** Same as Table 4, but for the comparison of ACE-FTS v3.5 and the Bruker 125HR partial columns (2007-2013).

| Species | $N$ | Altitude range (km) | Mean distance at 20 km (km) | Mean time difference (h) | Mean beta angle of ACE occultations (°) | PC diff (%) | $R$ | slope |
|---|---|---|---|---|---|---|---|---|
| $O_3$ | 95 | 9.0–48.5 | 601 | 3.9 | 27.2 | $3.6 \pm 0.6$ | 0.91 | $0.92 \pm 0.04$ |
| HCl | 94 | 9.0–39.0 | 604 | 3.9 | 27.2 | $2.4 \pm 0.6$ | 0.92 | $0.98 \pm 0.04$ |
| $HNO_3$ | 91 | 9.0–30.5 | 599 | 3.9 | 25.8 | $1.5 \pm 1.0$ | 0.77 | $0.82 \pm 0.05$ |
| HF | 104 | 14.0–39.0 | 602 | 3.9 | 26.3 | $-1.9 \pm 1.0$ | 0.84 | $0.91 \pm 0.05$ |

**Table 6.** Same as Table 4, but for the comparison of the tropospheric species for ACE-FTS v3.5 and PARIS-IR partial columns (2006-2013). Different coincidence criteria are used. Compared here is the maximum distance between the observed air masses at 14 km along the line-of-sight for both instruments.

| Species | Distance criterion | $N$ | Altitude range (km) | Mean distance at 14 km (km) | Mean time difference (h) | Mean beta angle of ACE occultations (°) | PC diff (%) | R | slope |
|---|---|---|---|---|---|---|---|---|---|
| $CH_4$ | 1000 km | 188 | 8.0–41.5 | 638 | 3.7 | 29.9 | $2.7 \pm 0.2$ | 0.64 | $1.02 \pm 0.06$ |
| $CH_4$ | 500 km | 65 | 8.0–41.5 | 399 | 3.2 | 34.0 | $3.0 \pm 0.3$ | 0.78 | $0.97 \pm 0.07$ |
| $N_2O$ | 1000 km | 147 | 8.0–37.5 | 638 | 3.8 | 27.9 | $6.1 \pm 0.4$ | 0.62 | $1.04 \pm 0.06$ |
| $N_2O$ | 500 km | 59 | 8.0–37.5 | 400 | 3.4 | 31.5 | $6.6 \pm 0.5$ | 0.80 | $1.05 \pm 0.08$ |



**Table 7.** Same as Table 6, but for the comparison of ACE-FTS v3.5 and the Bruker 125HR partial columns (2007-2013).

| Species | Distance criterion | $N$ | Altitude range (km) | Mean distance at 14 km (km) | Mean time difference (h) | Mean beta angle of ACE occultations (°) | PC diff (%) | R | slope |
|---|---|---|---|---|---|---|---|---|---|
| $CH_4$ | 1000 km | 112 | 6.5–34.0 | 647 | 3.9 | 31.3 | $0.5 \pm 0.3$ | 0.68 | $0.75 \pm 0.05$ |
| $CH_4$ | 500 km | 39 | 6.5–34.0 | 409 | 3.8 | 33.5 | $0.6 \pm 0.4$ | 0.89 | $0.79 \pm 0.06$ |
| $N_2O$ | 1000 km | 106 | 6.5–22.0 | 645 | 4.0 | 29.6 | $-1.6 \pm 0.4$ | 0.67 | $0.69 \pm 0.05$ |
| $N_2O$ | 500 km | 37 | 6.5–22.0 | 416 | 3.9 | 31.8 | $-1.6 \pm 0.5$ | 0.84 | $0.73 \pm 0.06$ |
| CO | 1000 km | 166 | 9.0–48.5 | 632 | 3.9 | 28.7 | $10.9 \pm 1.3$ | 0.68 | $0.89 \pm 0.05$ |
| CO | 500 km | 55 | 9.0–48.5 | 403 | 3.5 | 32.3 | $7.1 \pm 1.8$ | 0.80 | $0.86 \pm 0.07$ |
| $C_2H_6$ | 1000 km | 85 | 8.0–19.5 | 625 | 3.6 | 31.4 | $27.4 \pm 3.9$ | 0.60 | $0.87 \pm 0.08$ |
| $C_2H_6$ | 500 km | 33 | 8.0–19.5 | 413 | 3.8 | 32.2 | $20.6 \pm 5.5$ | 0.75 | $0.76 \pm 0.09$ |





**Figure 1.** Example of a total column averaging kernel (AVK, solid lines) and sensitivity (dashed-dotted lines) of the retrieval are shown for each of the eight species for PARIS-IR (red) and the Bruker 125HR (blue). If no dashed-dotted line is shown in the figure, the sensitivity and total column averaging kernel are too similar to distinguish the difference on the plot.







**Figure 2.** PARIS-IR versus Bruker 125HR total columns for each of the trace gases used in this study, showing the correlation before (cyan triangles) and after smoothing (red dots). The line of best fit is shown as a thin grey line for the unsmoothed total columns and as a thick black line for the smoothed total columns. The dashed black line represents the 1-to-1 line as a reference. Slopes and correlation coefficients are given in Table 3.




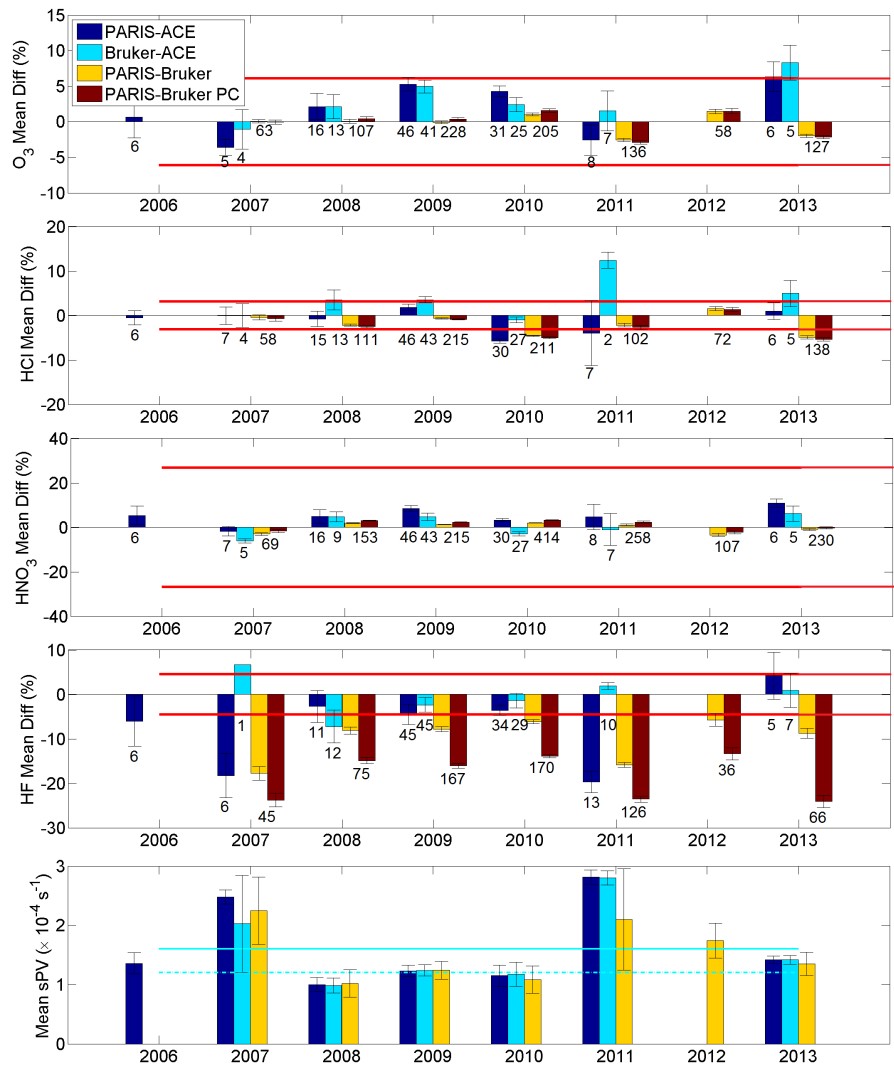

**Figure 3.** Mean differences for the stratospheric species estimated for each year (late February to early April) between PARIS-IR and ACE-FTS (blue), the Bruker 125HR and ACE-FTS (cyan) and between PARIS-IR and the Bruker 125HR using total columns (yellow) and partial columns (red) for the comparison. The error bars display the standard error of the mean differences. The number displayed above each bar represents the number of pairs. The combined retrieval uncertainty from PARIS-IR and the Bruker 125HR is shown as a red line. The lower panel illustrates the average sPV along the line-of-sight at 20 km of the pairs that are compared. The cyan solid and dashed lines represent the inner and outer polar vortex edge, respectively.





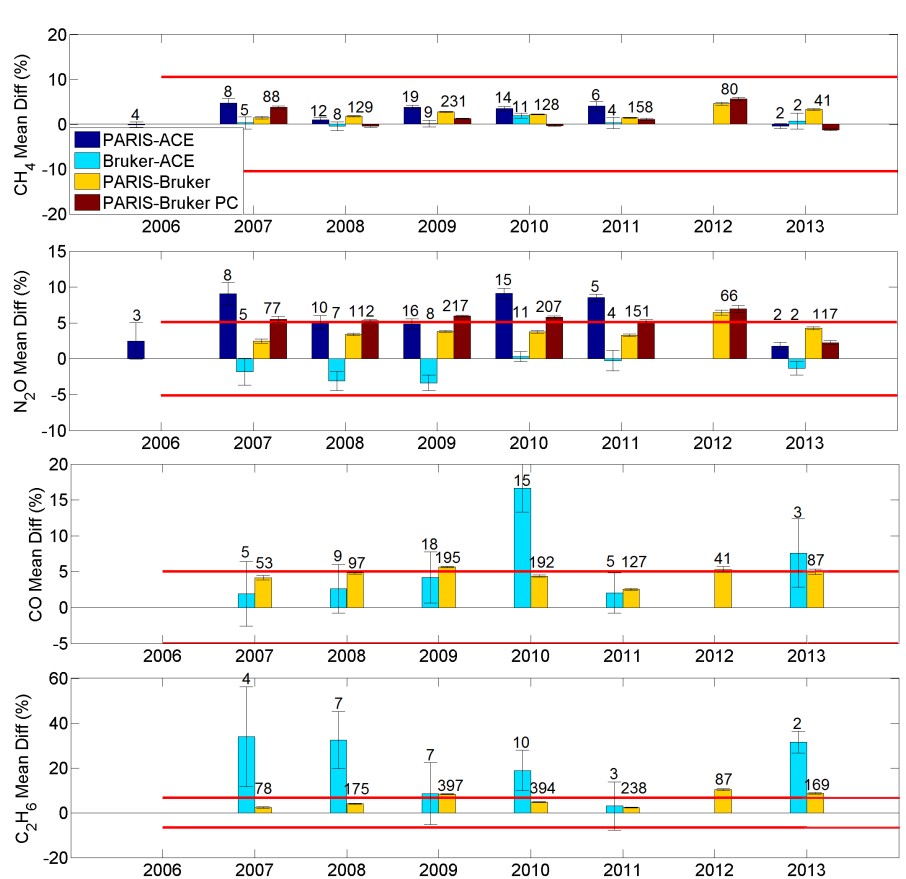

**Figure 4.** Same as Fig. 3 but for the tropospheric species.





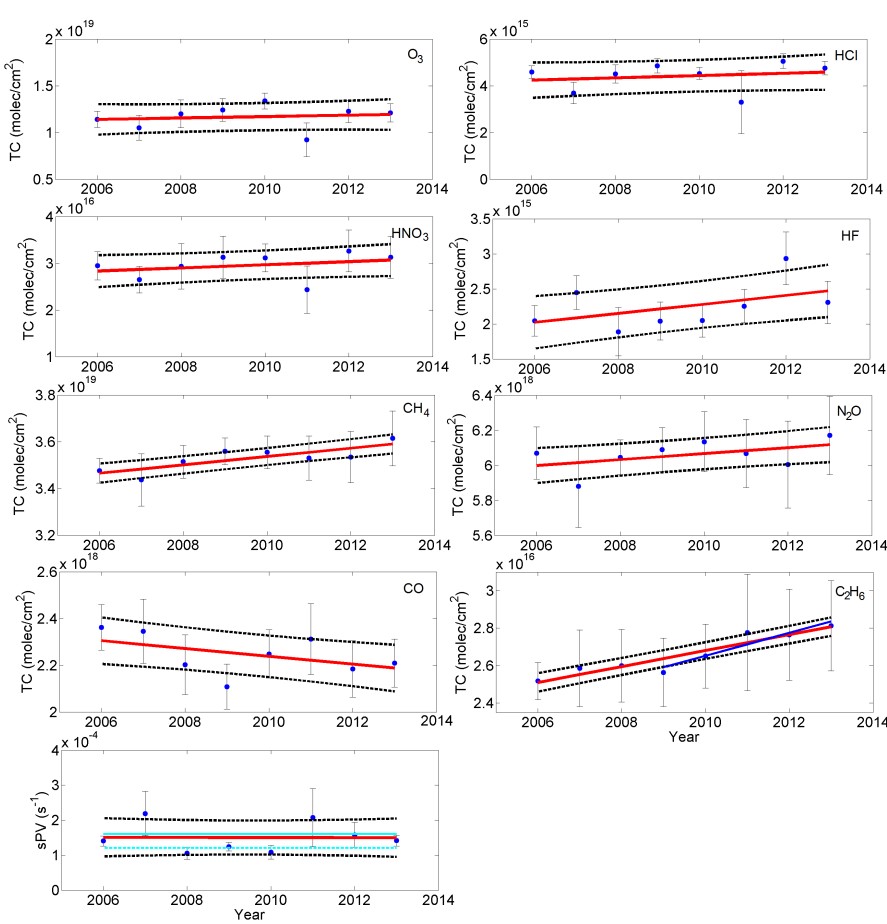

**Figure 5.** Yearly springtime campaign averages (blue dots) and $1\sigma$ standard deviation (error bar) between 2006 and 2013 obtained from the PARIS-IR total column retrievals for all eight trace gases used in this study. The sPV at $20\,\text{km}$ along the line-of-sight is shown in the lower panel, together with the inner (solid cyan line) and outer (dashed cyan line) edge of the polar vortex. The red solid lines represent the lines of best fit and the black dashed lines display the $1\sigma$ standard deviation of the fit. The best fit between 2009 and 2013 is displayed for $C_2H_6$ as a dark blue line.





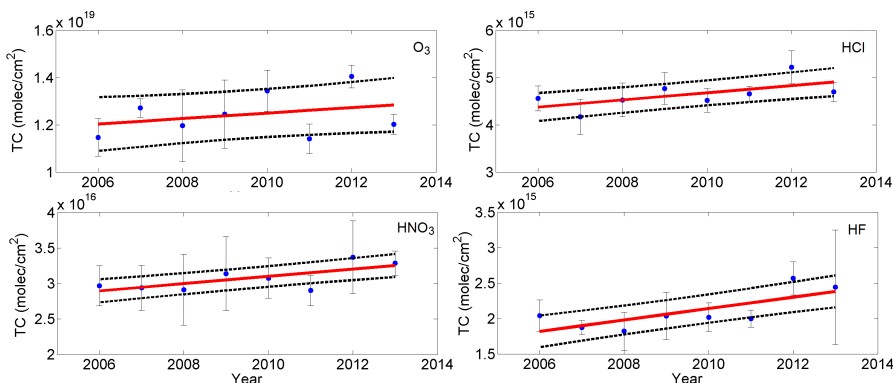

**Figure 6.** Springtime campaign averages (and $1\sigma$ standard deviation) between 2006 and 2013 obtained from the PARIS-IR total column retrievals for all stratospheric species for measurements that were taken outside the polar vortex (sPV$< 1.2 \times 10^{-4}\,\mathrm{s}^{-1}$ at 20 km along the line-of-sight). The colour and symbol scheme is the same as in Fig. 5.