# Peer review of "Multi-year comparisons of ground-based and space-borne Fourier Transform Spectrometers in the high Arctic between 2006 and 2013"

_Atmospheric Measurement Techniques, 2016_

## Referee Comment (RC1) · Anonymous Referee #1 · 25 Oct 2016

A useful paper which clearly fits in the scope of AMT. Well organized an well written. I have just a couple of issues wrt language and method.

Method:

Sect 2: I understand that VMR is retrieved directly and I assume that you would have mentioned if one of the retrievals was in the log domain. If you have used direct VMR fits then everything is ok. If log(VMR) is retrieved for some instruments, then it should be mentioned and might add some complication in the inter-comparison and interpretation of averaging kernels etc.

p7 l33 - p8 l2. I am a bit confused because I thought that the formalism for error

estimation in Rodgers 1976 where the error is directly inferred from the information matrix automatically includes the smoothing error (Eq 18 in Rodgers 1976). In contrast, in Rodgers 1990 the retrieval noise and the smoothing error (called null space error in Rodgers 1990) are evaluated separately. To my understanding only the Rodgers 1990 formalism allows to evaluate the pure noise without any smoothing error component (see also Eqs. 3.19 and 3.29-3.31 in the Rodgers 2000 book; 3.31 seems to be the one reported in Rodgers 1976, and it includes the smoothing error). Please clarify which error estimation formalism has been used and verify that the smoothing error has really not been included, not even implicitly via Eq 18 in Rodgers (1976).

p10 l22 and elsewhere: To judge how significant a correlation coefficient larger than 0.95 actually is, it would be necessary to also report the sample size along with the value. By the way, I suggest to mention somewhere that R is the correlation coefficient.

p10 l29 and elsewhere: There is a trap in comparing smoothed higher-resolution profiles with coarser resolved profiles. The application of the averaging kernel (Eq. 1) has considered also for the error of the better resolved profile (i.e. multiply the groundbased error covariance matrix from the left and the right (transposed) by the averaging kernel, $(S_{new} = A S_{old} A^T$ , smoothing typically makes the errors smaller). Without consideration of the error propagation through the smoothing process, the conclusion from the comparison will be too optimistic. Please check if this propagation has been considered. This is relevant to all conclusions where the combined retrieval uncertainties are mentioned. Please verify that this error propagation of the smoothing operation is considered in the error estimates used, and mention this, because this is often forgotten.

Language:

p2 l24: "next" is ambiguous. I think it is typically understood as the one after the current (i.e. in this case the intro) but is used here for the one after the last mentioned. Perhaps "following" might be clearer.

p5 l20: "take approximately every 7 min". Here it is not quite clear to me if this is the

time for a single spectrum or for the resulting spectrum after co-adding.

p10 l15: I am always confused how the word DOFS is correctly used, particularly if DOFS are "large" or "many". My intuition says me that the "number of degrees..." is "large" but that the "degrees..." are "many" but I may be wrong.

———————————————————

---

## Referee Comment (RC2) · Anonymous Referee #3 · 8 Nov 2016

This paper describes comparisons between multi-year trace gas retrieval datasets from two ground-based Fourier Transform Spectrometers located in the high Arctic. The ground-based retrievals are also compared to satellite retrievals from the ACE-FTS (v3.5). The trace gases are split into "stratospheric" (O3, HCl, HNO3, HF) and "tropospheric" (CH4, N2O, CO, C2H6) species for the purposes of the analysis. The authors use the longer of the two ground-based datasets to assess long-term variation/increases of these gases.

Overall, these results represent an interesting long-term dataset. The validation of the ACE-FTS v3.5 retrievals also has value. The paper is appropriate for AMT. The methodology is sound (although I have a few questions in places – see below). I

recommend that this work be published after minor revisions.

Major comments

The paper describes comparisons between the ground-based radiometers and the ACE-FTS v3.5 retrievals. When broken down by year, the number of co-incidences seems rather too small to provide a rigorous validation of the satellite products, especially for the so-called tropospheric species. The abstract states that there is no significant increase in the mean differences over the eight years of the comparison. "Significant" has meaning in a statistical sense. Statistically, what magnitude of change would be significant, given the numbers of points involved in the comparisons?

Are there other studies that have been used to assess the stability of the ACE-FTS retrievals over time? If so, they should be cited here. If not, then this should be stated.

I would have liked to have seen a scatter plot that shows the ground-based retrievals for all years compared to the ACE-FTS retrievals for all years, perhaps with different colors for measurements inside and outside the vortex. Is there a reason why such a figure is not shown?

Figures 5 and 6 could use improvement. The molecular labels on the plots are hard to see and the plots appear to be quite low resolution.

The text could use some editing to prune redundant text, since there is a lot of repetition. Some of the figures need to be better quality. Specific comments are provided below.

Minor comments and typos/grammar

Page 2, line 8: PARIS-IR should be spelled out at the first mention in the abstract

Page 2, last sentence of the abstract: This is an odd sentence. "Increased (gases) near PEARL" makes it sound as though there is some kind of spatial gradient, and stating that these are observed in the PARIS-IR implies somehow that the increases were not

observed in the Bruker dataset. Suggest changing this to read something more like, "Increases in (gases) have been observed in the PARIS-IR dataset, the longer of the two ground-based records."

Page 3, Lines 1-3: There is something not right about the grammar in this sentence.

Page 3, Line 18: Does this paper really focus on the retrieval of partial and total column values, or should you rather say that you focus on the analysis of the retrievals?

Page 3, line 22: CO and C2H6 also have industrial sources. C2H6 is also associated with oil and natural gas extraction. CO also has a non-negligible biogenic source, which is becoming an increasingly large component of the overall budget as anthropogenic emissions decrease in the developed world (see, for example, Hudman etla., GRL, 2008). These also ought to be mentioned here, if you state biomass burning as a source. If you are going to be consistent, you might also state the sources for CH4 and N2O. Since this is not a paper focused on sources, perhaps you don't need to spend space talking about sources, but to only mention biomass burning without mentioning other sources could be misleading.

Page 3, lines 30-34: Lists of numbers in the text are difficult to read and I am not convinced that listing all the numbers here is instructive. There is no information here that would let the reader know whether these numbers can be compared directly to the numbers in the Tables in this work or not. If it makes sense to compare these numbers to the numbers from this work, then they should be listed in a table, not in the text. If the numbers listed here from previous work cannot be directly compared to the numbers from this work, say, because they are from different latitudes and/or use different coincidence criteria, then there doesn't seem much point in listing them. It would perhaps be more instructive to have a table with these numbers for the ACE v2.2 comparisons, with some brief description of the latitude regime, number of cases etc involved. Alternatively, you might choose not to show the numbers, but just to discuss the important points about what has and has not been done in terms of validation of

both the v2.2 and v3.5 ACE-FTS datasets, and how this study adds to the existing body of knowledge.

Page 4, lines 2-4: Again, this list of numbers is tough to follow and I am not sure what to conclude from it. Are these the numbers that are most directly comparable to the numbers in the Tables in this work?

Page 4, lines 10-13: See comments above. Why list % differences for v2.2 validation but not for the v3.5 studies that you cite here? A consistent approach is needed.

Page 4, line 15: The wording should be updated here. You are not performing a comparison of multiple trace gases, you are performing a comparison between the satellite and ground-based data for multiple trace gases.

Page 4, line 19: Again, care should be taken with the wording. Strictly speaking, the method and criteria do not reduce the biases.

Page 5, lines7-8: "1/3 of the beam is directed into PARIS-IR and 2/3 of the beam into the Bruker". For those not familiar with the instrumental set-up – what does this mean?

Page 5, lines 8-9: "During the campaign, the satellite-based ACE-FTS took measurements near Eureka..." The way that this sentence is written implies that the ACE-FTS took special observations for the purpose of the campaign, targeted for the Eureka location. Was this the case, or was the satellite just making the observations that it would have made regardless of whether or not there was instrumentation on the ground at that location at that time? The same question applies to the way this is worded in page 8, lines 12-13.

Page 5, lines 19-20: The wording here is unclear. Does it take 7 minutes to acquire 20 spectra? Also, the last sentence in this paragraph seems redundant.

Page 5, line 30: "depending on the filter range" – This is unclear. Are the filter ranges changed from time to time, or does the "two or four co-added spectra" depend on the filter range? (Some filter ranges require four co-added spectra, due to the instrument

noise characteristics, while other ranges with lower noise need only two?)

Page 6, lines 25-26 and page 7/8: Why use a daily profile, rather than using the profiles from the different model time steps? I did not understand the description of the estimation of temperature errors in the retrieval. What do you mean, "averaged radiosondes"? Does this approach account for errors due to variability of the temperature profile with time? Please provide further explanation.

End of page 6/start of page 7: "A forward model is used to generate a model atmosphere from this a priori information..." What does this mean? In my mind, a forward model usually refers to the calculation of radiances, given the input atmospheric state.

Page 7, line 24: "the total column averaging kernel is forced to 1 at all altitudes." I don't know what this means and what the justification is. Please provide some further explanation.

Page 9, line 24: Technically, smoothing does not improve the intercomparisons. Consider rephrasing this to just say instead that accounting for the difference in vertical resolution between the two instruments is necessary in order to assess biases between the retrievals.

Section 3.2: Why use 0.5 x (PARIS + Bruker)? Why not just use one of the instruments as the reference? Also, there is a lot of repetition within this section as well as repetition of material from previous sections. This could use some editing.

Page 10, line 23: "no significant bias". How do you determine what is significant here? Presumably, the significance of the bias should be somehow related to the atmospheric variability of each gas, and probably also to what you might want to use the retrievals for? For example, $CH_4$ is significantly less variable than CO. What will these ground-based trace gas retrievals be used for, aside from ACE-FTS validation?

End of Section 3.2: Are you saying that you conclude that SFIT4 provides more accurate results than SFIT2? If so, what would be the likely reasons?

Section 4: There is a lot of repetition in this section. Given that is was expected that a tighter coincidence criterion would result in closer agreement between ground-based and satellite measurements, a lot of space is devoted to this issue in both 4.1 and 4.2. This could be shortened considerably.

Page 13, lines 21-34: If all these numbers are in the Tables, is there a need to list them here in the text? It is much easier to look at a table.

Page 14, lines 10-14: The explanation of partial/total column differences for HF is not very clear. Is the underlying issue here that the a priori profile used for HF is on the low side, but the observed values tend to be enhanced? Please clarify.

Page 15, line 31: What about the C2H6? Could the difference in C2H6 be due to plumes seen by the ground-based instrument but not by ACE-FTS? Have you looked to see whether the enhancements in CO observed from the ground-based instruments were coincident with enhancements in C2H6?

End of page 16, start of page 17: "Thus, there is the potential for a bias...". I did not understand the two sentences a the end of this paragraph. Please find a way to rephrase to make the point clear.

Acknowledgements: What is DMP? Is this spelled out anywhere?

---

## Author Comment (AC1) · 29 Apr 2017

We would like to thank reviewer #1 for his/her corrections and recommendations.

**Sect 2: I understand that VMR is retrieved directly and I assume that you would have mentioned if one of the retrievals was in the log domain. If you have used direct VMR fits then everything is ok. If log(VMR) is retrieved for some instruments, then it should be mentioned and might add some complication in the inter-comparison and interpretation of averaging kernels etc.**
Yes, the VMR was retrieved directly for the ground-based instruments and the space-borne ACE-FTS and none of these used a log domain. It was therefore not mentioned

in the text.

**p7 l33 - p8 l2. I am a bit confused because I thought that the formalism for error estimation in Rodgers 1976 where the error is directly inferred from the information matrix automatically includes the smoothing error (Eq 18 in Rodgers 1976). In contrast, in Rodgers 1990 the retrieval noise and the smoothing error (called null space error in Rodgers 1990) are evaluated separately. To my understanding only the Rodgers 1990 formalism allows to evaluate the pure noise without any smoothing error component (see also Eqs. 3.19 and 3.29-3.31 in the Rodgers 2000 book; 3.31 seems to be the one reported in Rodgers 1976, and it includes the smoothing error). Please clarify which error estimation formalism has been used and verify that the smoothing error has really not been included, not even implicitly via Eq 18 in Rodgers (1976).**
This has been corrected and the reference has been updated to Rodgers (2000) instead of Rodgers (1976). The smoothing error is not included in our error estimate.

**p10 l22 and elsewhere: To judge how significant a correlation coefficient larger than 0.95 actually is, it would be necessary to also report the sample size along with the value. By the way, I suggest to mention somewhere that R is the correlation coefficient.**
We have changed the sentence to:
"The correlation is excellent for $O_3$, HCl, $HNO_3$, and CO, with correlation coefficients $R \geq 0.95$ and the slopes of the regression plot between 0.93 and 1.13 ($N = 685$ to 1623), see Table 3."

Throughout the text, the term "correlation coefficient" has been added each time $R$ was mentioned.

**p10 l29 and elsewhere: There is a trap in comparing smoothed higher-resolution profiles with coarser resolved profiles. The application of the averaging kernel (Eq. 1) has considered also for the error of the better resolved profile (i.e. multiply the groundbased error covariance matrix from the left and the right (transposed) by the averaging kernel, (Snew = ASoldAT , smoothing typically makes the errors smaller). Without consideration of the error propagation through the smoothing process, the conclusion from the comparison will be too optimistic. Please check if this propagation has been considered. This is relevant to all conclusions where the combined retrieval uncertainties are mentioned. Please verify that this error propagation of the smoothing operation is considered in the error estimates used, and mention this, because this is often forgotten.**

The error after smoothing has not been estimated in our study since the error covariance matrix is not available for all instruments used in this study, and a consistent approach has been used throughout the paper.

**Language:**

**p2 l24: "next" is ambiguous. I think it is typically understood as the one after the current (i.e. in this case the intro) but is used here for the one after the last mentioned. Perhaps "following" might be clearer.**

p4, l. 24, we have changed the wording of the sentence according to the suggestion:
"The following section focuses on the methodology and results of the ACE-FTS comparison results."

**p5 l20: "take approximately every 7 min". Here it is not quite clear to me if this**

**is the time for a single spectrum or for the resulting spectrum after co-adding.**
We have changed the wording of the sentence:
"Each measurement is recorded approximately every 7 min and consists of 20 co-added spectra (Sung et al., 2007)."

**p10 l15: I am always confused how the word DOFS is correctly used, particularly if DOFS are "large" or "many". My intuition says me that the "number of degrees..." is "large" but that the "degrees..." are "many" but I may be wrong.**
We have changed the wording of the sentence to the suggested one:
"The differences between the smoothed and unsmoothed columns are relatively large $\sim 9\,\%$ for HF, for which the total column retrievals from PARIS-IR have DOFS of approximately 1, whereas the Bruker 125HR HF retrievals have twice as many DOFS (see Table 1)."

---

## Author Comment (AC2) · 29 Apr 2017

We would like to thank reviewer #3, for his/her corrections and recommendations.

**Major comments**

The paper describes comparisons between the ground-based radiometers and the ACE-FTS v3.5 retrievals. When broken down by year, the number of co-incidences seems rather too small to provide a rigorous validation of the satellite products, especially for the so-called tropospheric species. The

**abstract states that there is no significant increase in the mean differences over the eight years of the comparison. "Significant" has meaning in a statistical sense. Statistically, what magnitude of change would be significant, given the numbers of points involved in the comparisons?**

The text in the abstract has been edited to:

"The comparisons show no notable increases of the mean differences over these eight years,..."

**Are there other studies that have been used to assess the stability of the ACE-FTS retrievals over time? If so, they should be cited here. If not, then this should be stated.**

There are no other publications that have assessed the stability of the ACE-FTS dataset over time. The following sentence has been included on p.4 l.15 to clarify this:

"For the first time, the stability of the ACE-FTS dataset is examined over an eight year time period. "

**I would have liked to have seen a scatter plot that shows the ground-based retrievals for all years compared to the ACE-FTS retrievals for all years, perhaps with different colors for measurements inside and outside the vortex. Is there a reason why such a figure is not shown?**

We have now provided four additional figures as supplementary figures, showing scatter plots for the ACE-FTS and PARIS-IR as well as the ACE-FTS and Bruker 125HR comparisons. The measurements taken inside the polar vortex, near the edge and outside the polar vortex are highlighted for the stratospheric species. The text has been edited on p.13 l.16:

"Scatter plots of the partial column comparisons between the ACE-FTS and ground-based datasets for the stratospheric species can be found in the supplementary material (Figs. S1 and S2)."

And on p.15 l.7:
"Scatter plots of the partial column comparisons between the ACE-FTS and ground-based datasets for the tropospheric species can be found in the supplementary material (Figs. S3 and S4)."

**Figures 5 and 6 could use improvement. The molecular labels on the plots are hard to see and the plots appear to be quite low resolution.**
Figures 5 and 6 have been changed, the molecular labels have a larger font size now, and the plots appear in better resolution.

**The text could use some editing to prune redundant text, since there is a lot of repetition. Some of the figures need to be better quality. Specific comments are provided below.**
These have been addressed and the text and some figures have been edited, see specific comments below.

**Minor comments and typos/grammar**

**Page 2, line 8: PARIS-IR should be spelled out at the first mention in the abstract**
This has been addressed and the acronym definition now appears on page 1, l. 8/9

**Page 2, last sentence of the abstract: This is an odd sentence. "Increased (gases) near PEARL" makes it sound as though there is some kind of spatial gradient, and stating that these are observed in the PARIS-IR implies somehow that the increases were not observed in the Bruker dataset. Suggest changing this to read something more like, "Increases in (gases) have been observed in the PARIS-IR dataset, the longer of the two ground-based records."**
The sentence now reads:

[Figure]

"Increased $O_3$ ($0.9\,\%\mathrm{yr}^{-1}$), HCl ($1.7\,\%\mathrm{yr}^{-1}$), HF ($3.8\,\%\mathrm{yr}^{-1}$), $CH_4$ ($0.5\,\%\mathrm{yr}^{-1}$) and $C_2H_6$ ($2.3\,\%\mathrm{yr}^{-1}$, 2009-2013) have been found with the PARIS-IR dataset, the longer of the two ground-based records."

**Page 3, Lines 1-3: There is something not right about the grammar in this sentence.**
The sentence has been changed to:
"These ground-based FTS datasets extend over a long time period and capture many species, thus contributing to the ongoing validation of the satellite-based instrument and helping assess whether ACE-FTS measurements have remained consistent over the last decade."

**Page 3, Line 18: Does this paper really focus on the retrieval of partial and total column values, or should you rather say that you focus on the analysis of the retrievals?**
This sentence now reads:
"Herein, we focus on the analysis of retrieved partial and total column values, derived from infrared FTS spectra, for $O_3$ together with several molecules important in catalytic $O_3$ destruction."

**Page 3, line 22: CO and C2H6 also have industrial sources. C2H6 is also associated with oil and natural gas extraction. CO also has a non-negligible biogenic source, which is becoming an increasingly large component of the overall budget as anthropogenic emissions decrease in the developed world (see, for example, Hudman etla., GRL, 2008). These also ought to be mentioned here, if you state biomass burning as a source. If you are going to be consistent, you might also state the sources for CH4 and N2O. Since this is not a paper focused on sources, perhaps you don't need to spend space talking**

**about sources, but to only mention biomass burning without mentioning other sources could be misleading.**
These sentences have been removed.

**Page 3, lines 30-34: Lists of numbers in the text are difficult to read and I am not convinced that listing all the numbers here is instructive. There is no information here that would let the reader know whether these numbers can be compared directly to the numbers in the Tables in this work or not. If it makes sense to compare these numbers to the numbers from this work, then they should be listed in a table, not in the text. If the numbers listed here from previous work cannot be directly compared to the numbers from this work, say, because they are from different latitudes and/or use different coincidence criteria, then there doesn't seem much point in listing them. It would perhaps be more instructive to have a table with these numbers for the ACE v2.2 comparisons, with some brief description of the latitude regime, number of cases etc involved. Alternatively, you might choose not to show the numbers, but just to discuss the important points about what has and has not been done in terms of validation of both the v2.2 and v3.5 ACE-FTS datasets, and how this study adds to the existing body of knowledge.**
**Page 4, lines 2-4: Again, this list of numbers is tough to follow and I am not sure what to conclude from it. Are these the numbers that are most directly comparable to the numbers in the Tables in this work?**
This paragraph has been edited:
"The earlier ACE-FTS retrieval version, ACE-FTS v2.2+updates, of these trace gases has previously been validated for most species discussed in this study, ($O_3$ (Dupuy et al., 2009), HCl (Mahieu et al., 2008), $HNO_3$ (Wolff et al., 2008), HF (Mahieu et al., 2008), $CH_4$ (DeMazière et al., 2008), $N_2O$ (Strong et al., 2008), CO (Clerbaux et al., 2008)). In these studies, partial column comparisons between ground-based FTSs and ACE-FTS in the Arctic typically show larger differences than comparisons at

lower latitudes. The inclusion of criteria ensuring that similar air masses are sampled with respect to the polar vortex reduces this difference seen in the Arctic significantly. Using these additional criteria, Batchelor et al. (2010) and Fu et al. (2011) found much improved mean differences, between the ACE-FTS (v2.2+updates) and ground-based FTS datasets, for $O_3$, HCl, $HNO_3$, and HF comparisons by approximately a factor of 2. These mean differences are comparable to the typical differences at lower latitudes. "

**Page 4, lines 10-13: See comments above. Why list % differences for v2.2 validation but not for the v3.5 studies that you cite here? A consistent approach is needed.**
The previous paragraph has been edited and now excludes specific percent differences.

**Page 4, line 15: The wording should be updated here. You are not performing a comparison of multiple trace gases, you are performing a comparison between the satellite and ground-based data for multiple trace gases.**
This sentence now reads:
"A comprehensive comparison between ACE-FTS v3.5 and ground-based FTS measurements of multiple trace gases is provided, including measurements that were taken inside and outside the polar vortex."

**Page 4, line 19: Again, care should be taken with the wording. Strictly speaking, the method and criteria do not reduce the biases.**
This sentence now reads:
"For these comparisons, we will use the same method and criteria for the viewing geometry as Batchelor et al. (2010) and Fu et al. (2011), which have been shown to improve the comparison between ground- and satellite-based instruments in the Arctic."

**Page 5, lines7-8: "1/3 of the beam is directed into PARIS-IR and 2/3 of the beam into the Bruker". For those not familiar with the instrumental set-up - what does this mean?**
The exact details of the beam sharing are not important in the scope of this study and the sentence has been changed to:
"These two instruments are located side-by-side in the PEARL Ridge Laboratory and share a solar beam from the same sun tracker installed on the roof above."

**Page 5, lines 8-9: "During the campaign, the satellite-based ACE-FTS took measurements near Eureka: : :" The way that this sentence is written implies that the ACE-FTS took special observations for the purpose of the campaign, targeted for the Eureka location. Was this the case, or was the satellite just making the observations that it would have made regardless of whether or not there was instrumentation on the ground at that location at that time? The same question applies to the way this is worded in page 8, lines 12-13.**
ACE-FTS takes solar occultation measurements, therefore, targeted measurements are not possible. Every year during polar spring the satellite takes occultation measurements in the high Arctic. This sentence in the paper has been changed to clarify this:
"During the campaigns, the satellite-based ACE-FTS took routine measurements in the high Arctic and provided profiles of over 30 trace gases. Details of these instruments and their datasets are given below."
And the sentence on p. 7/8 has been edited:
"Its mission goals include improving our understanding of polar ozone chemistry, thus every year during the Arctic sunrise period, ACE takes measurements over the high Arctic, near Eureka."

**Page 5, lines 19-20: The wording here is unclear. Does it take 7 minutes to**

**acquire 20 spectra? Also, the last sentence in this paragraph seems redundant.**
As also noted in the response to reviewer #1, this sentence has been changed to:
"Each measurement is recorded approximately every 7 min and consists of 20 co-added spectra (Sung et al., 2007)."

**Page 5, line 30: "depending on the filter range" - This is unclear. Are the filter ranges changed from time to time, or does the "two or four co-added spectra" depend on the filter range? (Some filter ranges require four co-added spectra, due to the instrument noise characteristics, while other ranges with lower noise need only two?)**
This sentence now reads:
"Therefore, subject to favourable weather conditions and depending on the filter sequence, each species is measured approximately every 30 min."

**Page 6, lines 25-26 and page 7/8: Why use a daily profile, rather than using the profiles from the different model time steps? I did not understand the description of the estimation of temperature errors in the retrieval. What do you mean, "averaged radiosondes"? Does this approach account for errors due to variability of the temperature profile with time? Please provide further explanation.**
The change in temperature throughout the day is accounted for in the error analysis. Temperature and pressure differences that occur within one day impact the retrieval by less than 1%. Furthermore, this approach of using a daily temperature profile (from NCEP) is also in alignment with the current recommendations from NDACC IRWG for the retrievals of atmospheric species from ground-based mid-IR spectra.
Radiosondes are launched twice a day in Eureka, as such these have been averaged to a "daily" temperature profile and then compared to the daily NCEP temperature profiles. This has been clarified in the text, p.7 l.28:

"The random temperature error (Table 2) is based on the difference between the NCEP temperature profiles and "measured profiles" created by averaging the twice-daily radiosonde profiles. The systematic temperature error is based on the NCEP temperature error profile (Table 2)."

**End of page 6/start of page 7: "A forward model is used to generate a model atmosphere from this a priori information: : :" What does this mean? In my mind, a forward model usually refers to the calculation of radiances, given the input atmospheric state.**
The text has been edited:
"The forward model used in SFIT is a radiative transfer model that is utilized to generate a modelled absorption spectrum from this a priori information based on the daily pressure and temperature information, as well as the location of the measurement site."

**Page 7, line 24: "the total column averaging kernel is forced to 1 at all altitudes." I don't know what this means and what the justification is. Please provide some further explanation.**
The different retrieval method used for the Bruker $CH_4$ retrieval is the Tikhonov method. This is not an essential part of this study and as such is only briefly mentioned here. Further details can be found in Sussmann et al. (2011). The sentence has been changed to:
"Because a different retrieval technique has been used to determine the Bruker 125HR $CH_4$ (Sussmann et al., 2011), the total column averaging kernel (as shown in Fig. 1) is 1 at all altitudes."

**Page 9, line 24: Technically, smoothing does not improve the intercomparisons. Consider rephrasing this to just say instead that accounting for the difference in vertical resolution between the two instruments is necessary in order to assess**

**biases between the retrievals.**
We have changed the sentence to:
"The different resolutions are accounted for here by smoothing the VMR profiles following the method described in (Rodgers and Connor, 2003). Accounting for the difference in vertical resolution between the two instruments has been addressed in numerous publications (e.g., Batchelor et al., 2010; Griffin et al., 2013)."

**Section 3.2: Why use 0.5 x (PARIS + Bruker)? Why not just use one of the instruments as the reference? Also, there is a lot of repetition within this section as well as repetition of material from previous sections. This could use some editing.**
To avoid the assumption that one retrieval is the correct one, we use the mean as a reference. This approach is also consistent to previous studies that have done similar instrument comparisons (e.g., Clerbaux et al., 2008; Batchelor et al, 2010; Fu et al., 2011).
Section 3.2 has been edited to avoid repetition.

**Page 10, line 23: "no significant bias". How do you determine what is significant here? Presumably, the significance of the bias should be somehow related to the atmospheric variability of each gas, and probably also to what you might want to use the retrievals for? For example, CH4 is significantly less variable than CO. What will these groundbased trace gas retrievals be used for, aside from ACE-FTS validation?**
The statement has been removed:
"The correlation is excellent for $O_3$, HCl, $HNO_3$, and CO, with correlation coefficients $R \geq 0.95$ and the slopes of the regression plot between 0.93 and 1.13 when between 685 and 1623 coincident measurements are compared (see Table 1)."

**End of Section 3.2: Are you saying that you conclude that SFIT4 provides more accurate results than SFIT2? If so, what would be the likely reasons?**
We are saying that the differences are comparable (or slightly better for some species) with the differences found in previous studies (that compared similar datasets, however, not exactly the same one). Further investigation and a more direct comparison between SFIT2 and SFIT4 (with the exact dataset) would be required to conclude with certainty that SFIT4 is more accurate than SFIT2. This was not the focus of this study.

**Section 4: There is a lot of repetition in this section. Given that is was expected that a tighter coincidence criterion would result in closer agreement between ground-based and satellite measurements, a lot of space is devoted to this issue in both 4.1 and 4.2. This could be shortened considerably.**
The text in Section 4.1/4.2 has been edited and shortened to avoid repetition. We also edited these paragraphs to clarify and highlight our results.

**Page 13, lines 21-34: If all these numbers are in the Tables, is there a need to list them here in the text? It is much easier to look at a table.**
The text has been edited and now excludes these numbers that are already listed in the tables. Also, p.15 l.9-24 has been edited in a consistent way.

**Page 14, lines 10-14: The explanation of partial/total column differences for HF is not very clear. Is the underlying issue here that the a priori profile used for HF is on the low side, but the observed values tend to be enhanced? Please clarify.**
We have clarified this paragraph as follows:
"This is due to the low vertical resolution of the PARIS-IR HF retrieval, for which the partial columns have generally less than 1 DOFS ($\sim 0.8$) and are therefore more strongly influenced by the a priori profile. The impact of the total column versus the partial column comparison is not as significant for $O_3$, HCl, and $HNO_3$, where the
partial columns contain more information from the measurement."

**Page 15, line 31: What about the C2H6? Could the difference in C2H6 be due to plumes seen by the ground-based instrument but not by ACE-FTS? Have you looked to see whether the enhancements in CO observed from the ground-based instruments were coincident with enhancements in C2H6?**

We did investigate whether the enhancement could be from a plume, but we did not see simultaneous enhancements of $C_2H_6$ and CO in our ground-based dataset.

**End of page 16, start of page 17: "Thus, there is the potential for a bias: : :". I did not understand the two sentences a the end of this paragraph. Please find a way to rephrase to make the point clear.**
These sentences have been changed to:
"Therefore, the airmass outside the polar vortex has been measured for a few days in early March 2007, and in late March to early April 2011. Thus, there is the potential for a temporal sampling influence as measurements in 2007 and 2011 are taken in different months, approximately one month apart."

**Acknowledgements: What is DMP? Is this spelled out anywhere?**
This sentence now reads:
"We are also very grateful to William Daffer from JPL, who carried out the Derived Meteorological Parameters calculations."

Please also note the supplement to this comment:
http://www.atmos-meas-tech-discuss.net/amt-2016-272/amt-2016-272-AC2-supplement.pdf

**[AMTD](AMTD)**

Interactive
comment

[Figure]

[Figure]

**Supplement:**

**Multi-year comparisons of ground-based and space-borne Fourier Transform Spectrometers in the high Arctic between 2006 and 2013**

**Debora Griffin et al.**

*Correspondence to:* Kaley A. Walker
(kaley.walker@utoronto.ca)

This document contains four supplementary figures that show correlation plots of the partial column comparison between ACE-FTS and the ground-based FTSs, PARIS-IR and the Bruker 125HR.

5

[Figure]

**Figure S1.** PARIS-IR partial columns versus ACE-FTS partial columns for each of the stratospheric trace gases used in this study, showing the correlation after smoothing has been applied to the ACE-FTS profiles. Measurements taken inside, near the edge and outside the polar vortex are shown as red dots, cyan triangles, and blue triangles, respectively. The line of best fit is shown as a black line. The dashed black line represents the 1-to-1 line as a reference. Slopes and correlation coefficients are given in Table 4.

[Figure]

**Figure S2.** Same as Fig. S1, but for the Bruker 125HR partial columns versus smoothed ACE-FTS partial columns. Slopes and correlation coefficients are given in Table 5.

[Figure]

**Figure S3.** PARIS-IR partial columns versus ACE-FTS partial columns (red dots) for each of the tropospheric trace gases used in this study, showing the correlation after smoothing has been applied to the ACE-FTS profiles. The line of best fit is shown as a black line. The dashed black line represents the 1-to-1 line as a reference. Slopes and correlation coefficients are given in Table 6.

[Figure]

**Figure S4.** Same as Fig. S3, but for the Bruker 125HR partial columns versus smoothed ACE-FTS partial columns. Slopes and correlation coefficients are given in Table 7.